# Utilizing Earth Observations of Soil Freeze/Thaw Data and Atmospheric Concentrations to Estimate Cold Season Methane Emissions in the Northern High Latitudes

Maria Tenkanen [1,*] , Aki Tsuruta [1] , Kimmo Rautiainen [2] , Vilma Kangasaho [1] , Raymond Ellul [3] and Tuula Aalto [1]

1   Climate System Research, Finnish Meteorological Institute, 00560 Helsinki, Finland; Aki.Tsuruta@fmi.fi (A.T.); Vilma.Kangasaho@fmi.fi (V.K.); Tuula.Aalto@fmi.fi (T.A.)
2   Earth Observation Research, Finnish Meteorological Institute, 00560 Helsinki, Finland; Kimmo.Rautiainen@fmi.fi
3   Atmospheric Research, Department of Geosciences, University of Malta, MSD 2080 Msida, Malta; ray.ellul@um.edu.mt
*   Correspondence: Maria.Tenkanen@fmi.fi

**Abstract:** The northern wetland methane emission estimates have large uncertainties. Inversion models are a qualified method to estimate the methane fluxes and emissions in northern latitudes but when atmospheric observations are sparse, the models are only as good as their a priori estimates. Thus, improving a priori estimates is a competent way to reduce uncertainties and enhance emission estimates in the sparsely sampled regions. Here, we use a novel way to integrate remote sensing soil freeze/thaw (F/T) status from SMOS satellite to better capture the seasonality of methane emissions in the northern high latitude. The SMOS F/T data provide daily information of soil freezing state in the northern latitudes, and in this study, the data is used to define the cold season in the high latitudes and, thus, improve our knowledge of the seasonal cycle of biospheric methane fluxes. The SMOS F/T data is implemented to LPX-Bern DYPTOP model estimates and the modified fluxes are used as a biospheric a priori in the inversion model CarbonTracker Europe-CH₄. The implementation of the SMOS F/T soil state is shown to be beneficial in improving the inversion model's cold season biospheric flux estimates. Our results show that cold season biospheric CH₄ emissions in northern high latitudes are approximately 0.60 Tg lower than previously estimated, which corresponds to 17% reduction in the cold season biospheric emissions. This reduction is partly compensated by increased anthropogenic emissions in the same area (0.23 Tg), and the results also indicates that the anthropogenic emissions could have even larger contribution in cold season than estimated here.

**Keywords:** methane flux; methane emissions; biospheric flux; northern high latitudes; atmospheric inversion; cold season; soils

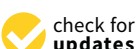



## 1. Introduction

In the northern high latitudes (NHL), there are vast wetland areas where over a quarter of the total soil carbon is stored [1,2]. In addition to storing a large amount of carbon, these wetlands are a large source of methane (CH₄), which is the second most important anthropogenic greenhouse gas in terms of the radiative forcing [3]. The magnitude and seasonal cycle of NHL wetlands' methane emissions are, however, highly uncertain [4,5], and climate change will likely affect ecosystem dynamics in the NHL wetlands, which could increase the methane emissions [6]. Especially, permafrost thawing threatens the carbon stock, which has been primarily remained intact and sequestered from the atmosphere until now [7–10].

A well-known characteristic of NHL wetland methane emissions is their strong seasonal cycle. During summer, methane fluxes are high when soil is moist and warm. In autumn, the methane fluxes start to decrease when soil temperature also decreases, and

when the top-most layer of soil freezes, the methane fluxes are expected to reach their minimum [11,12]. Most methane is emitted in summer but low winter fluxes do not necessarily mean that they are insignificant in terms of annual and regional budgets. A cold season can be in some regions as long as half of the year and, thus, the cold season methane emissions might add up and may be a notable part of the annual emission budgets [13,14]. Previous studies based on in situ observations have also shown large methane bursts at the start and end of the cold season when soil is freezing or thawing [15–18].

Periodic soil freezing and thawing strongly affect the seasonality of NHL wetland emissions. However, the effect is complicated and, thus, hard to model. The current land-surface models included in the Coupled Model Intercomparison Project Phase 6 (CMIP6) earth system models still lack many of the needed properties [19]. One main shortage is the depth of the model's soil profile. Many models do not have deep enough soils to represent the seasonally thawed active layer on top of the permafrost. There are also questions on how insulations of snow [20,21] and vegetation (mainly moss and lichen etc.) [22,23] should be modelled. Furthermore, the spatial extent of snow coverage and vegetation is hard to observe on a large scale and, thus, hard to ascertain that the models represent those correctly. All these variables also affect methane production and the release of methane from soil. Here, we concentrate on the cold season, which is defined by the freezing and thawing of soil and for this purpose, it is important to define the period of frozen soil as accurately as possible.

The NHL cold season methane emission estimates have more uncertainties than growing season emissions because there are still many challenges. In situ methane flux measurements rarely extend over the cold season due to technical difficulties caused by snow and ice, and the availability of satellite atmospheric column retrievals, which use sunlight, is limited in high latitudes during the cold season due to polar nights.

We use a remote sensing soil state product called the Soil Moisture and Ocean Salinity Freeze/Thaw (SMOS F/T) soil state [24] and quantify the cold season in the northern latitudes as the period when soil is frozen to better capture the seasonality of methane emissions in the NHL. The SMOS F/T soil state informs whether the soil is frozen or thawed, and it extends also over the cold season in the NHL. SMOS is the European Space agency's mission that is part of Living Planet Programme [25]. It is the first earth-observing satellite, which measures the passive microwave L-band (1.4 GHz) globally and its measurements are used to retrieve several variables related to soil moisture and ocean surface salinity, which are two key variables within the Earth's water cycle. SMOS's low operating frequency allows to retrieve the state from deeper soil, and it is much less affected by the overlying vegetation than the previously used high-frequency remote sensing instruments [25,26].

Other data available during the NHL cold season, which enables us to estimate $CH_4$ fluxes, are atmospheric mole fraction observations. The mole fraction data can be used in atmospheric inversion models to estimate the methane fluxes spatially and temporally. The inversion models assimilate the measurements with a combination of a transport model and a priori knowledge of the fluxes. The reliability of the optimised fluxes is largely dependent on these three major components (measurements, a transport model and a priori fluxes), so to improve the inversion, we should improve at least one of them, and here we concentrate on the a priori fluxes.

We improve biospheric methane flux estimates of the dynamic process model LPX-Bern DYPTOP [27] by defining the cold season more accurately with the SMOS F/T soil state. The improved flux estimates are further optimised by the atmospheric inversion model CarbonTracker Europe-$CH_4$ (CTE-CH4) [28] for years 2010–2018. As a result, we obtain novel estimates for cold season methane emissions in the northern high latitudes.

## 2. Materials and Methods

### 2.1. Definitions

Throughout the study, we use certain definitions, which describe different regions and time periods. The definitions and the reasoning behind them are listed below.

*NHL*  Latitudes above 50°N.
    In our analyses, we take the NHL as the land area above 50°N. The land area is defined according to the TransCom regions [29].

*NHL sites* Observations stations above 49°N latitude.
    We also address NHL observation stations. We decided to include observation stations above 49°N latitude, a criterion differing from the NHL definitions because there are six stations near the Canadian-U.S. border below the 50°N latitudinal zone, which we wanted to include.

*Wintertime* Time period when the soil is frozen.
    We use the term "wintertime" as the period when soil is frozen according to the SMOS F/T soil state. This is defined individually for each $1° \times 1°$ (latitude $\times$ longitude) grid cell.

*Cold season* November to April.
    To unify the wintertime results meaningfully, we defined the "cold season" based on the individual grid cells' "wintertime". We calculated the median freezing and thawing dates each year above the latitude 50°N and took the mean of the days. The median freezing day was November 6. and the median thawing day April 30. Thus, we defined the "cold season" as the time period from November to April.

### 2.2. The SMOS Soil Freeze/Thaw

The SMOS F/T (version 2.000) service [24] gives daily values of soil freezing states in the Northern Hemisphere. The SMOS F/T data are available since July 2010 and it is gridded on EASE Grid 2.0 [30] with a resolution of 25 km $\times$ 25 km. The SMOS F/T data are available online from ESA SMOS dissemination server [31] and from FMI [32].

SMOS is the European Space Agency's Soil Moisture and Ocean Salinity mission [25]. SMOS satellite measures the passive microwave radiation on the L-band (1.4 GHz). The measured radiation originates from approximately 5–15 cm below soil's surface and, thus, the information it withholds is truly from soil itself. This has a clear advantage over the older similar freeze/thaw product the FT Earth system data record (FT ESDR) [33]. FT ESDR uses high frequency (37 GHz) remote sensing data meaning the information is mostly limited only to vegetation or snow layer. Passive microwave radiation is not dependent on sunlight, which enables us to define soil properties in the northern high latitudes during winter when sunlight's availability is limited.

Soil's brightness temperature (temperature of a black body that is in thermal equilibrium with its surroundings) is used to derive the soil's freezing state. The brightness temperature can be derived from passive microwave radiation. It correlates with the soil's dielectric constant, which is different for liquid water and ice on SMOS' observed frequency. When soil freezes, its dielectric constant decreases, which is seen as an increase in the brightness temperature. When soil thaws, the situation is vice versa, i.e., the dielectric constant increases and the brightness temperature decreases. This relation between soil freezing/thawing and the brightness temperature is the basis of the SMOS F/T soil state algorithm since we can measure the brightness temperature and with it, define whether the source is frozen or thawed [24].

The SMOS F/T algorithm is based on threshold detection. Each SMOS observation is compared pixel-wise to frozen and thawed ground references and categorised into three categorical levels as "frozen", "partially frozen" or "thawed". The references have been built up based on the observations from 2010 to 2018. For this purpose SMOS F/T algorithm uses two auxiliary datasets: ECMWF's re-analyses 2 m air temperature (ERA-Interim [34] until the end of July 2018 and CAMS NRT (Copernicus Atmosphere Monitoring Service

Near-real-time) after that) and IMS snow cover data [35]. The auxiliary datasets are also used in making a processing mask, which detects obvious erroneous soil F/T estimates. The obvious erroneous soil F/T estimates can occur during summer when a false frozen soil detection is estimated. If the daily average air temperature has stayed above zero centigrade since June, the F/T estimate is forced to stay in the "thawed" category. Another correction to the SMOS F/T estimates due to the processing mask can occur during winter. If there are severe winter conditions (no above zero centigrade for 10 consecutive days and average air temperature below −3 °C), the F/T estimate is not allowed to change to the "thawed" category. Note, this does not force the soil estimate to the "frozen" category, it only does not allow soil thawing in such pixels that have already been estimated to be frozen. It is possible, that the soil stays non-frozen even though there are severe winter conditions if, for example, the snow layer is thick.

*2.3. CarbonTracker Europe-CH$_4$*

CarbonTracker Europe-CH$_4$ is an atmospheric inverse model that estimates surface methane fluxes globally [28,36]. It uses in situ atmospheric methane mole fraction observations to constrain a priori fluxes from biospheric and anthropogenic sources simultaneously. The fluxes are optimised based on the ensemble Kalman filter method [37] with ensemble size of 500 and 5-week lag.

Within CTE-CH$_4$, we used TM5 [38] as an observation operator to transform the methane fluxes to atmospheric mole fractions. TM5 is run at global 4° × 6° (latitude × longitude) horizontal resolution with 1° × 1° zoom and 2° × 3° intermediate zoom surrounding the 1° × 1° zoom grid over Europe (see, e.g., [39]), constrained by 3-hourly ECMWF ERA-Interim meteorological data. We acknowledge that ECMWF has a new reanalysis product, ERA5, and it is currently being implemented for the model simulations. The monthly atmospheric sink due to photochemical reactions with OH, Cl, and O($^1$D) was taken into account based on Houweling et al. (2014) [40] and Brühl and Crutzen (1993) [41]. The interannual variability of the atmospheric sink was not considered, and the atmospheric sink was not optimised in this study.

CH$_4$ fluxes were constrained with atmospheric methane mole fraction data mainly from the Integrated Carbon Observation System (ICOS), National Oceanic and Atmospheric Administration/Global Monitoring Laboratory (NOAA/GML) ObsPack v2.0 [42] and World Data Centre for Greenhouse Gases (WDCGG [43]) that contained data from global in situ stations. Additionally, we added the NHL observations from in situ stations of the Finnish Meteorological Institute (FMI), National Institute for Environmental Studies (NIES), and Max Planck Institute for Biogeochemistry (MPI-Biogeochemistry). The data contained both weekly discrete air samples and hourly continuous measurements. The hourly data were preprocessed before inversion by taking daily averages similarly to Tsuruta et al. (2017) [28]. The observation uncertainties were defined for each site and observation based on site characteristics and measurement accuracy. The observation uncertainty also included information about TM5's ability to reproduce the mole fractions. Those values are somewhat arbitrary, but those are based on previous studies [28,29,44], and reflects the ability of the model to forecast the atmospheric concentrations well.

During the assimilation process, if the absolute differences between observations and prior mole fractions were larger than 30 ppb, the observations were rejected, i.e., not assimilated to constrain the fluxes. This rejection criterion was slightly different from the previous CTE-CH$_4$ version (e.g., [28]), but it did not affect much the number of assimilated observations in the NHL sites. The assimilation rate for the study period overall sites was >99%. Sites and their uncertainties are listed in Table 1 (NHL sites) and Table S1 in the Supplementary Materials (global sites excluding the NHL sites). The NHL site locations are shown in Figure 3 and daily averaged NHL observations are in the Supplementary Materials Figure S1.

**Table 1.** List of surface observation sites used in inversions, located above 49°N. Observation Uncertainty (Obs. Unc.) is used to define diagonal values in the observation covariance matrix. Data type is categorized into two measurements—Discrete (D) and Continuous (C).

| Sitecode | Site Name | Country | Contributor | Longitude (°E) | Latitude (°N) | Height * (m a.s.l.) | Obs. Unc. (ppb) | Data Type (D/C) | Date min. ** (Year/Month) | Date max. (Year/Month) |
|---|---|---|---|---|---|---|---|---|---|---|
| ABT | Abbotsford, British Columbia | Canada | Environment and Climate Change Canada (ECCC) | −122.34 | 49.01 | 93 | 30.0 | C | 2014/03 | 2018/12 |
| ALT | Alert, Nunavut | Canada | NOAA/GML | −62.51 | 82.45 | 195 | 15.0 | D | 1998/01 | 2018/12 |
| ALT | Alert, Nunavut | Canada | ECCC | −62.51 | 82.45 | 210 | 15.0 | C | 1998/01 | 2018/12 |
| AZV | Azovo | Russian Federation | NIES | 73.03 | 54.71 | 190 | 30.0 | C | 2009/10 | 2017/12 |
| BAL | Baltic Sea | Poland | NOAA/GML | 17.22 | 55.35 | 28 | 75.0 | D | 1998/01 | 2011/06 |
| BAR | Baranova | Russian Federation | FMI | 101.62 | 79.28 | 30 | 4.5 | C | 2015/11 | 2019/12 |
| BCK | Behchoko, Northwest Territories | Canada | ECCC | −115.92 | 62.80 | 220 | 15.0 | C | 2010/10 | 2018/12 |
| BIR | Birkenes | Norway | Norwegian Institute for Air Research (NILU) | 8.25 | 58.39 | 219 | 25.0 | C | 2009/05 | 2018/12 |
| BLK | Baker Lake, Nunavut | Canada | ECCC | −96.01 | 64.33 | 61 | 15.0 | C | 2017/07 | 2018/12 |
| BRA | Bratt's Lake Saskatchewan | Canada | ECCC | −104.71 | 50.20 | 630 | 75.0 | C | 2009/10 | 2018/12 |
| BRW | Barrow Atmospheric Baseline Observatory | United States | NOAA/GML | −156.61 | 71.32 | 27.5 | 15.0 | C | 1998/01 | 2018/12 |
| BRW | Barrow Atmospheric Baseline Observatory | United States | NOAA/GML | −156.61 | 71.32 | 16.0, 11.0, 13.0, 27.5 | 15.0 | D | 1998/01 | 2018/12 |
| CBA | Cold Bay, Alaska | United States | NOAA/GML | −162.72 | 55.21 | 57.04, 25.0 | 15.0 | D | 1998/01 | 2018/12 |
| CBW | Cabaw | Netherlands | Netherlands Organisation for Applied Scientific Research (TNO) | 4.93 | 51.97 | 199 | 15.0 | C | 2005/01 | 2013/06 |
| CBY | Cambridge Bay, Nunavut Territory | Canada | ECCC | −105.06 | 69.13 | 47 | 15.0 | C | 2012/12 | 2018/12 |
| CHL | Churchill, Manitoba | Canada | ECCC | −93.82 | 58.74 | 89 | 15.0 | C | 2011/10 | 2018/12 |
| CHM | Chibougamau, Quebec | Canada | ECCC | −74.34 | 49.69 | 423 | 30.0 | C | 2007/08 | 2011/04 |
| CHS | Cherskii | Russian Federation | NOAA/GML | 161.53 | 68.51 | 64.4 | 25.0 | C | 2008/07 | 2016/01 |
| CPS | Chapais,Quebec | Canada | ECCC | −74.98 | 49.82 | 386.0, 399.0, 431.0 | 15.0 | C | 2011/12 | 2018/12 |
| DEM | Demyanskoe | Russian Federation | NIES | 70.87 | 59.79 | 155 | 30.0 | C | 2005/09 | 2017/12 |
| ESP | Estevan Point, British Columbia | Canada | ECCC | −126.54 | 49.38 | 47 | 25.0 | C | 2009/03 | 2018/12 |
| EST | Esther, Alberta | Canada | ECCC | −110.21 | 51.67 | 710.0, 757.0 | 30.0 | C | 2010/01 | 2018/12 |
| ETL | East Trout Lake, Saskatchewan | Canada | ECCC | −104.99 | 54.35 | 598 | 30.0 | C | 2005/08 | 2018/12 |
| FNE | Fort Nelson, British Columbia | Canada | ECCC | −122.57 | 58.84 | 376 | 30.0 | C | 2014/07 | 2018/12 |

**Table 1.** *Cont.*

| Sitecode | Site Name | Country | Contributor | Longitude (°E)] | Latitude (°N)] | Height * (m a.s.l.) | Obs. Unc. (ppb) | Data Type (D/C) | Date min. ** (Year/Month) | Date max. (Year/Month) |
|---|---|---|---|---|---|---|---|---|---|---|
| FSD | Fraserdale | Canada | ECCC | −81.57 | 49.88 | 250 | 30.0 | C | 1999/01 | 2018/12 |
| GAT | Gartow | Germany | ICOS-ATC, HPB | 11.44 | 53.07 | 411 | 25.0 | C | 2016/05 | 2018/12 |
| HTM | Hyltemossa | Sweden | ICOS-ATC, LUND-CEC | 13.42 | 56.10 | 265 | 25.0 | C | 2017/04 | 2018/12 |
| ICE | Storhofdi, Vestmannaeyjar | Iceland | NOAA/GML | −20.29 | 63.40 | 121.7, 127.0 | 15.0 | D | 1998/01 | 2018/12 |
| IGR | Igrim | Russian Federation | NIES | 64.41 | 63.19 | 89 | 30.0 | C | 2005/04 | 2013/07 |
| INU | Inuvik, Northwest Territories | Canada | ECCC | −133.53 | 68.32 | 123 | 15.0 | C | 2012/02 | 2018/12 |
| KJN | Kjolnes | Norway | University of Exeter | 29.23 | 70.85 | 20 | 15.0 | C | 2013/10 | 2019/04 |
| KMP | Kumpula | Finland | FMI | 24.96 | 60.20 | 53 | 30.0 | C | 2010/01 | 2019/12 |
| KRE | Kresin u Pacova | Czech Republic | ICOS-ATC, CAS | 15.08 | 49.58 | 784 | 25.0 | C | 2017/04 | 2018/12 |
| KRS | Karasevoe | Russian Federation | NIES | 82.42 | 58.25 | 156 | 30.0 | C | 2004/09 | 2017/12 |
| LEB | Lethbridge | Canada | ECCC | −112.80 | 49.73 | 25 | 30.0 | C | 2016/10 | 2018/03 |
| LIN | Lindenberg | Germany | ICOS-ATC, HPB | 14.12 | 52.17 | 171 | 30.0 | C | 2015/10 | 2018/12 |
| LLB | Lac La Biche, Alberta | Canada | NOAA/GML | −112.47 | 54.95 | 588.0, 546.1 | 30.0 | D | 2008/01 | 2013/02 |
| LLB | Lac La Biche, Alberta | Canada | ECCC | −112.47 | 54.95 | 550.0, 590.0 | 30.0 | C | 2007/04 | 2018/12 |
| LUT | Lutjewad | Netherlands | ICOS-ATC, RUG | 6.35 | 53.40 | 61 | 25.0 | C | 2018/08 | 2018/12 |
| MHD | Mace Head, County Galway | Ireland | NOAA/GML | −9.90 | 53.33 | 26 | 25.0 | D | 1998/01 | 2018/12 |
| MHD | Mace Head | Ireland | Laboratoire des Sciences du Climat et de l'Environnement (LSCE) | −9.90 | 53.33 | 29 | 25.0 | C | 2010/01 | 2018/04 |
| NGL | Neuglobsow | Germany | Umweltbundesamt Germany/Federal Environmental Agency (UBA-Germany) | 13.03 | 53.17 | 68.4 | 75.0 | C | 1999/01 | 2013/12 |
| NOR | Norunda | Sweden | ICOS-ATC, LUND-CEC | 17.48 | 60.09 | 146 | 15.0 | C | 2017/04 | 2018/12 |
| NOY | Noyabrsk | Russian Federation | NIES | 75.78 | 63.43 | 188 | 30.0 | C | 2005/10 | 2017/12 |
| OXK | Ochsenkopf | Germany | NOAA/GML | 11.81 | 50.03 | 1172.0, 1185.0 | 30.0 | D | 2003/03 | 2018/12 |
| PAL | Pallas-Sammaltunturi, GAW Station | Finland | NOAA/GML | 24.12 | 67.97 | 570 | 15.0 | D | 2001/12 | 2018/12 |
| PAL | Pallas-Sammaltunturi, GAW Station | Finland | FMI | 24.12 | 67.97 | 570 | 15.0 | C | 2004/02 | 2017/12 |
| PAL | Pallas-Sammaltunturi, GAW Station | Finland | ICOS-ATC, FMI | 24.12 | 67.97 | 577 | 15.0 | C | 2017/09 | 2018/12 |
| PUI | Puijo | Finland | FMI | 27.66 | 62.91 | 84 | 30.0 | C | 2011/11 | 2019/12 |
| RGL | Ridge Hill | United Kingdom | University of Urbino | −2.54 | 52.00 | 294 | 25.0 | C | 2012/02 | 2017/12 |
| SHM | Shemya Island, Alaska | United States | NOAA/GML | 174.13 | 52.71 | 28 | 25.0 | D | 1998/01 | 2018/10 |
| SMR | Hyytiala | Finland | ICOS-ATC, UHELS | 24.29 | 61.85 | 306 | 25.0 | C | 2016/12 | 2018/12 |
| SOD | Sodankylä | Finland | FMI | 26.64 | 67.36 | 227 | 25.0 | C | 2012/01 | 2019/12 |



**Table 1.** *Cont.*

| Sitecode | Site Name | Country | Contributor | Longitude (°E) | Latitude (°N) | Height * (m a.s.l.) | Obs. Unc. (ppb) | Data Type (D/C) | Date min. ** (Year/Month) | Date max. (Year/Month) |
|---|---|---|---|---|---|---|---|---|---|---|
| SUM | Summit | Greenland | NOAA/GML | −38.42 | 72.60 | 3214.5 | 15.0 | D | 1998/01 | 2018/12 |
| SVB | Svartberget | Sweden | ICOS-ATC, SLU | 19.78 | 64.26 | 385 | 25.0 | C | 2017/06 | 2018/12 |
| TAC | Tacolneston | United Kingdom | NOAA/GML | 1.14 | 52.52 | 236 | 25.0 | D | 2014/06 | 2016/01 |
| TAC | Tacolneston | United Kingdom | University of Bristol | 1.14 | 52.52 | 241 | 25.0 | C | 2013/01 | 2017/12 |
| TER | Teriberka | Russian Federation | Main Geophysical Observatory (MGO) | 35.10 | 69.20 | 42 | 15.0 | D | 1999/01 | 2019/12 |
| TIK | Hydrometeorological Observatory of Tiksi | Russian Federation | NOAA/GML | 128.89 | 71.60 | 29 | 15.0 | D | 2011/08 | 2018/09 |
| TIK | Tiksi | Russian Federation | FMI | 128.89 | 71.60 | 29 | 15.0 | C | 2010/09 | 2019/12 |
| TOH | Torfhaus | Germany | ICOS-ATC, HPB | 10.54 | 51.81 | 948 | 25.0 | C | 2017/12 | 2018/12 |
| UTO | Uto | Finland | ICOS-ATC, FMI | 21.37 | 59.78 | 65 | 25.0 | C | 2018/03 | 2018/12 |
| VGN | Vaganovo | Russian Federation | NIES | 62.32 | 54.50 | 277 | 30.0 | C | 2008/06 | 2017/12 |
| YAK | Yakutsk | Russian Federation | NIES | 129.36 | 62.09 | 344 | 30.0 | C | 2007/09 | 2013/12 |
| ZEP | Ny-Alesund, Svalbard | Norway and Sweden | NOAA/GML | 11.89 | 78.91 | 479 | 15.0 | D | 1998/01 | 2018/12 |
| ZEP | Ny-Alesund, Svalbard | Norway and Sweden | ICOS-ATC, NILU | 11.89 | 78.91 | 489 | 15.0 | C | 2017/07 | 2018/12 |
| ZOT | Zottino | Russian Federation | MPI | 89.21 | 60.48 | 415 | 25.0 | C | 2009/05 | 2016/12 |
| ZOT | Zottino | Russian Federation | MPI | 89.21 | 60.48 | 415 | 15.0 | D | 2006/10 | 2013/06 |

* Sampling heights from which atmospheric $CH_4$ is sampled in TM5. ** Observations used in this study between 2010 and 2018.

A priori fluxes include biospheric, anthropogenic, fire, termites and ocean sources, and these were taken from process-based models and inventory data. For biospheric fluxes, we used the estimates from The Land surface Processes and eXchanges DYPTOP (LPX-Bern DYPTOP) model version 1.4 [27,45–47] and its modified version with soil freeze/thaw state implemented into it (see Section 2.4). LPX-Bern DYPTOP is a dynamic global vegetation model, and the configuration with Dynamical Peatland Model Based on TOPMODEL (DYPTOP) combines an inundation model with a model determining suitability for peatland growth conditions to simulate the peatland spatial distribution and temporal changes. LPX-Bern DYPTOP uses meteorological data from CRU-TS3.26 [48] as an input. The model is forced with 2-m air temperature, but soil temperatures in different soil layers are simulated. The air temperature is given monthly and interpolated to daily values with cubic spline interpolation. The soil temperatures are also updated daily. Monthly precipitation values are stochastically distributed to the number of wet days which is also taken from CRU-TS3.26. The precipitation is determined to be snow if the interpolated daily temperature is lower than $-2$ °C.

For other a priori sources, we used estimates from EDGAR v5.0 ([49,50]) for anthropogenic, GFED v4.1s [51] for fire, VISIT [52] for termites, and those calculated based on ECMWF data for ocean sources [28]. Average total methane emissions in the NHL (above 50°N) during the study period (2010–2018) were 63.7 Tg/year.

CTE-CH$_4$ optimises CH$_4$ fluxes from anthropogenic and biospheric sources simultaneously. The temporal optimisation resolution was weekly. Spatial resolution varied from $1° \times 1°$ to region-wise (Figure S2). The highest $1° \times 1°$ optimisation resolution is applied to three NHL wetland regions: Canadian Hudson Bay Lowlands, Northern European Boreal region and northeast Russia (note that the high-resolution optimisation regions differ from the zoom region in TM5). These regions were chosen based on the availability of atmospheric methane mole fraction observations, which are used to constrain the fluxes, and following our interest to study NHL wetland emissions. The fluxes from other sources are not optimised.

The anthropogenic and biospheric sources are assumed to be independent, and the uncertainty of the prior fluxes over land is assumed to be 80% and 20% over oceans. The spatial correlation length varies from 100 to 900 km between optimisation regions based on the grid or optimisation region size and observation density (see Supplementary Materials for details). For the three NHL wetland regions, where fluxes are optimised at $1° \times 1°$ resolution, the correlation length is 100 km. We apply no temporal correlation on ensemble members, but only to the mean states by taking averages from the previous three weeks [36,37].

The simulations were conducted for the years 2010–2018 with 2 years (2008–2009) spin-up with LPX as a biospheric a priori. The years were chosen based on when SMOS F/T data and atmospheric observations were available. We are aware those data are constantly updated and will be extended in future studies.

### 2.4. Implementing the SMOS F/T Soil State to LPX-Bern DYPTOP Fluxes

To improve the inversion's cold season estimates, the SMOS Freeze/Thaw soil state was implemented to the estimates of LPX-Bern DYPTOP and obtained fluxes were used as a biospheric a priori in the CTE-CH$_4$. Methane fluxes from NHL wetlands are at their minimum when soil is frozen and since the SMOS F/T soil state obtains information on whether soil is frozen or thawed, we can define whether the wetlands should emit methane more than the minimum or not. We defined the wintertime for each $1° \times 1°$ grid with the SMOS F/T data and modified the LPX fluxes to be the magnitude of the winter minimum during the defined wintertime. The original LPX-Bern DYPTOP estimates are called from now on LPX and estimates with the SMOS F/T state implemented as LPX FT. The optimised estimates with LPX as the biospheric a priori is called from now on CTE and the optimised estimates with LPX FT as CTE FT.

To use the SMOS F/T soil state in our inversion, it was first transformed into the same coordinate projection with LPX (from EASE Grid 25 × 25 km to 1° latitude × 1° longitude). The coordinate transformation was done by calculating the ratio of frozen SMOS F/T pixels to all pixels in each 1° × 1° grid cell. The wintertime was then defined as the period when the ratio of frozen SMOS F/T pixels within a CTE-CH$_4$ 1° × 1° grid cell was at least 90%. The number of SMOS F/T pixels in each 1° × 1° grid cell differs depending on the latitude but in a large part of the NHL (higher than 56°N), it is less than 10. Therefore, with the 90% threshold, all the SMOS soil status pixels had to be frozen to fulfil the wintertime condition.

The monthly averaged methane fluxes were calculated from LPX for each grid cell (including years 2010–2017). The last year of the study period (2018) was excluded because LPX had estimates only until 2017. The smallest monthly mean was then taken as the wintertime flux. Next, the LPX was modified so that if the wintertime condition was fulfilled according to the SMOS F/T data in a grid cell, the wintertime flux was assigned to the grid cell. LPX emissions estimates from 2017 was used also for 2018, both in the SMOS F/T implementation and in the model run with LPX.

## 3. Results

### 3.1. Defined Cold Season

The average day-of-year of the start and end of the wintertime and the average wintertime length for each grid cell is shown in Figure 1. Soil starts to freeze the earliest in northeast Russia, northern Canada and Alaska in late August and the latest in the southern border of Russia, southern Fennoscandia and the northeastern United States in mid-December. The soil thawing happens almost reversely starting from the USA and the southern border of Eurasia around the New Year and occurring the latest in northeast Russia and northern Canada in June. The shortest wintertime length is a couple of days (the southern border of the SMOS F/T soil state data) and the longest wintertime lengths are over 220 days (the northernmost part of Canada and Russia).

The soil freezing proceeds from East to West in Eurasia (Figure 1a). This follows the permafrost zones [53], in other words, soil with most permafrost freezes first and permafrost-free soil freezes last. In spring, the soil thaw does not have such strong motion in a West–East direction and is most prominent in the South–North direction (Figure 1b). In North America, the same kind of phenomenon, where soil with more permafrost freezes first, can be observed, although, the freezing gradient is from north to south rather than in the West–East direction.

We compared start and end days of the cold season defined with the SMOS F/T soil state to freezing and thawing days defined with LPX-Bern DYPTOP's monthly temperature data. LPX-Bern DYPTOP has soil layers up to 2 m (and padding beneath that) where the first three layers are 10 cm deep each. The SMOS F/T soil state is retrieved from 5 to 15 cm beneath the surface and, thus, we took the average of the two first layers from LPX-Bern DYPTOP. We then defined the freezing day and thawing days from LPX-Bern DYPTOP as the first day of the month when the soil temperature dropped below 0 °C (as LPX-Bern DYPTOP gives monthly values).

The differences at the start and the end of the cold season in 2013 are presented in Figure 2. The SMOS F/T soil state showed later soil freezing than LPX-Bern DYPTOP in almost all other areas than in WSL and HBL. In WSL, 33% of the pixels froze earlier in the SMOS F/T data than in LPX-Bern DYPTOP, and the largest difference was 48 days. In HBL, 10% of the pixels froze earlier in the SMOS F/T data than in LPX-Bern DYPTOP. Furthermore, the largest difference was smaller, 20 days. In our SMOS F/T implementation, later soil freezing did not affect the cold season emission much since methane emissions were likely already in their winter minimum when the effect of the SMOS F/T implementation occurred. In contrast, in regions where the SMOS F/T soil state showed earlier soil freezing than LPX-Bern DYPTOP, the effect of the SMOS F/T implementation on the cold season emissions was emphasised. At the end of the cold season, the SMOS F/T soil state showed earlier soil thawing throughout the NHL. In 2013, the median difference over the NHL was

−27 days, in WSL −38 days and in HBL −32 days. Thus, overall, the SMOS F/T soil state approximately showed a month earlier soil thawing than LPX-Bern DYPTOP. This again meant that the SMOS F/T implementation had a lesser impact on the cold season methane emissions. The reasons behind the differences at the start and the end of the cold season are discussed in Section 4.2.

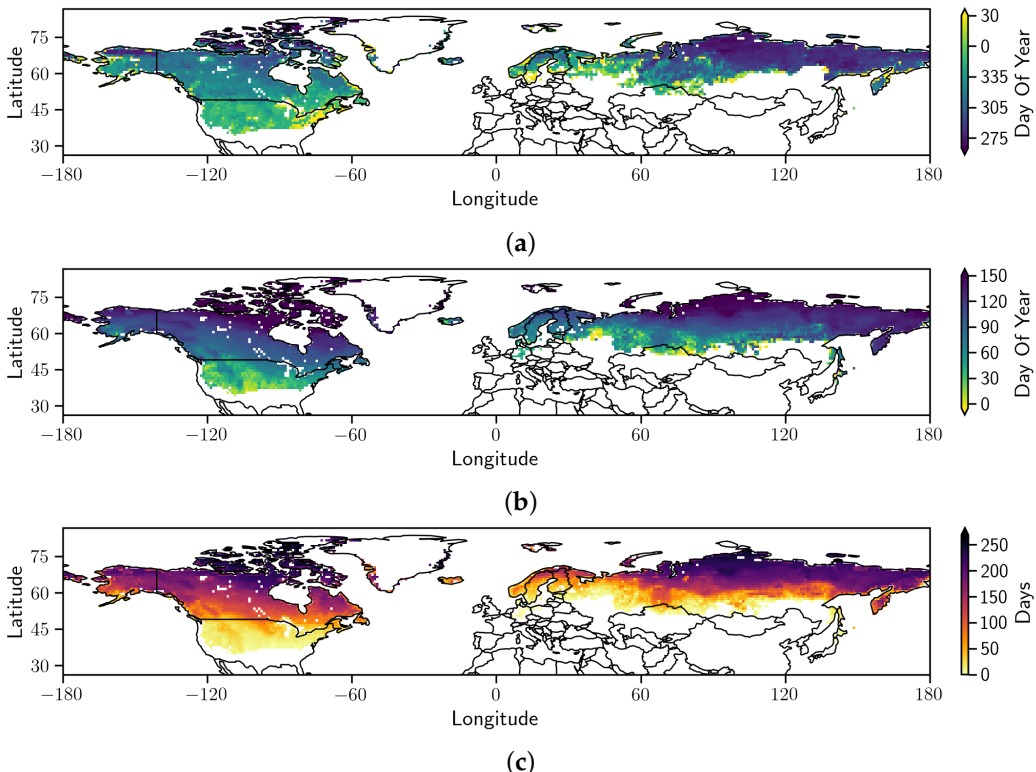

**Figure 1.** Average day-of-year when the wintertime (**a**) starts and (**b**) ends. (**c**) Average length of the wintertime in days. Values are defined based on the SMOS F/T soil state.

We decided to present the year 2013 here since the SMOS F/T soil state data coverage was abundant during 2013 and the difference in the start of the cold season between LPX-Bern DYPTOP and the SMOS F/T soil state can be seen clearly. We could also see similar differences in other years but the exact location varied annually (also depending on the data coverage). Thus, for example, the average difference in the start of the cold season over 2010–2018 was close to zero in WSL and HBL. The same applied also to the end of the cold season, although, there was less year-to-year variability, and the average over the study period was very similar to 2013.

### 3.2. Modified LPX Methane Emissions

The LPX biospheric methane emissions decreased by 0.80 Tg/year on average in the NHL when the SMOS F/T soil state was used to define the wintertime emissions. The LPX FT average annual methane emissions in the NHL were 16.7 Tg, and the cold season emissions were 3.2 Tg meaning the soil F/T implementation reduced the biospheric emissions by 4.8% annually and by 23% during the cold season.

The impact of SMOS F/T implementation on methane emission estimates was the most prominent in the Western Siberian Lowlands (WSL) (52–74°N, 60–94.5°E) and in Hudson Bay Lowlands (HBL) (50–60°N, 75–96°W) (regions shown in Figure 2). In WSL, the decrease was 0.27 Tg/year on average, i.e., 34% of the total emission decrease, and in HBL, the average annual decrease in the biospheric methane emissions was 0.12 Tg/year, i.e., 15% of the total decrease. Together WSL and HBL explained 49% of the total reduction in methane emissions.

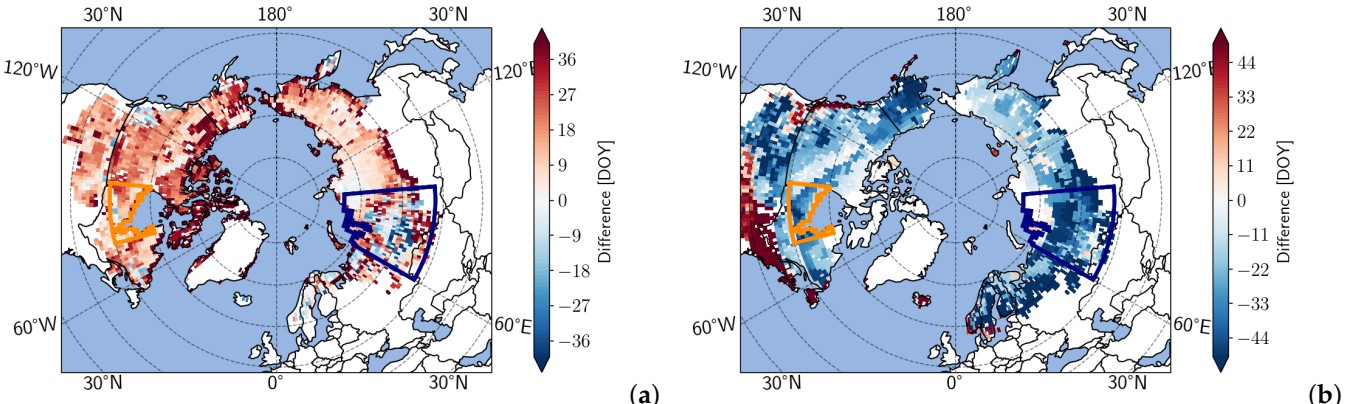

**Figure 2.** The difference of (**a**) the start and (**b**) the end of the cold season between the SMOS F/T soil state and ecosystem model LPX-Bern DYPTOP in 2013. For the SMOS F/T soil state, the values are given at daily resolution and for LPX-Bern DYPTOP those are at the monthly resolution. For LPX-Bern DYPTOP, the average of the first and the second soil layers is used. The average of the first and the second layers responds to the soil depth of 10 cm. Negative values mean that the SMOS F/T soil state showed earlier soil freezing (**a**) or thawing (**b**) than LPX-Bern DYPTOP, and positive values mean LPX-Bern DYPTOP froze/thawed earlier. The circulated areas are Hudson Bay Lowlands (orange) and Western Siberian Lowlands (blue).

### 3.3. Comparing Modelled and Observed Atmospheric Mole Fractions

Prior and posterior $CH_4$ atmospheric mole fractions were compared to in situ observations to study whether the modelled atmospheric mole fractions matched better with the observations when the SMOS F/T soil status was used. We calculated the detrended bias (model−observations) and Root Mean Square Error (RMSE) for 68 stations located above the latitude 49°N (the NHL sites). The SMOS F/T soil status implementation mainly affected the cold season fluxes so the statistics were calculated for the cold season, i.e., from November to April. We also investigated summer values (June–September) to see if there was an effect outside the cold season. Observations and model estimates were detrended using the ordinary least squares linear regression for the whole available dataset from each measurement station before the bias and RMSE calculations. From the detrended datasets, the bias was calculated as a mean of the difference between modelled and observed mole fractions. The difference between biases was then calculated as the difference between absolute values of two mean biases. For example, for the comparison between mole fractions from TM5 run with LPX or LPX FT as a biospheric a prior, the difference in the bias $\Delta_{\text{bias}}$ at one site was calculated as

$$\Delta_{\text{bias}} = |E(\mathbf{Y}_{\text{LPX}} - \mathbf{X})| - |E(\mathbf{Y}_{\text{LPX FT}} - \mathbf{X})|, \tag{1}$$

where $E$ denotes the mean, $\mathbf{Y}$ are modelled mole fraction, and $\mathbf{X}$ are observations. $\Delta_{\text{bias}}$ was positive when the absolute bias using LPX FT is smaller than that using LPX. This means that the bias using LPX FT is closer to zero than those using LPX, i.e., an improvement by the F/T implementation. Differences between RMSEs were calculated similarly. Terms cold season and summer bias and RMSE used in the article refer to these detrended values. The measurement availability varied from a couple of months to several years (median number of cold season days was 326). Sites where the data covered only a couple of months did not show any biases compared to sites with longer time series.

When the cold season biases and RMSEs without the SMOS F/T soil state implementation and with it and before the data assimilation (difference between mole fractions modelled with TM5 using LPX or LPX FT as the biospheric a priori) were compared, the statistics show an overall improvement. The median improvement in the cold season bias was 0.3 ppb and in the cold season RMSE 0.1 ppb. Furthermore, the median summer bias was reduced by 0.3 ppb and summer RMSE by 0.2 ppb. The differences in the cold season bias and RMSE between the prior mole fractions are shown in Figure 3 for each NHL site.

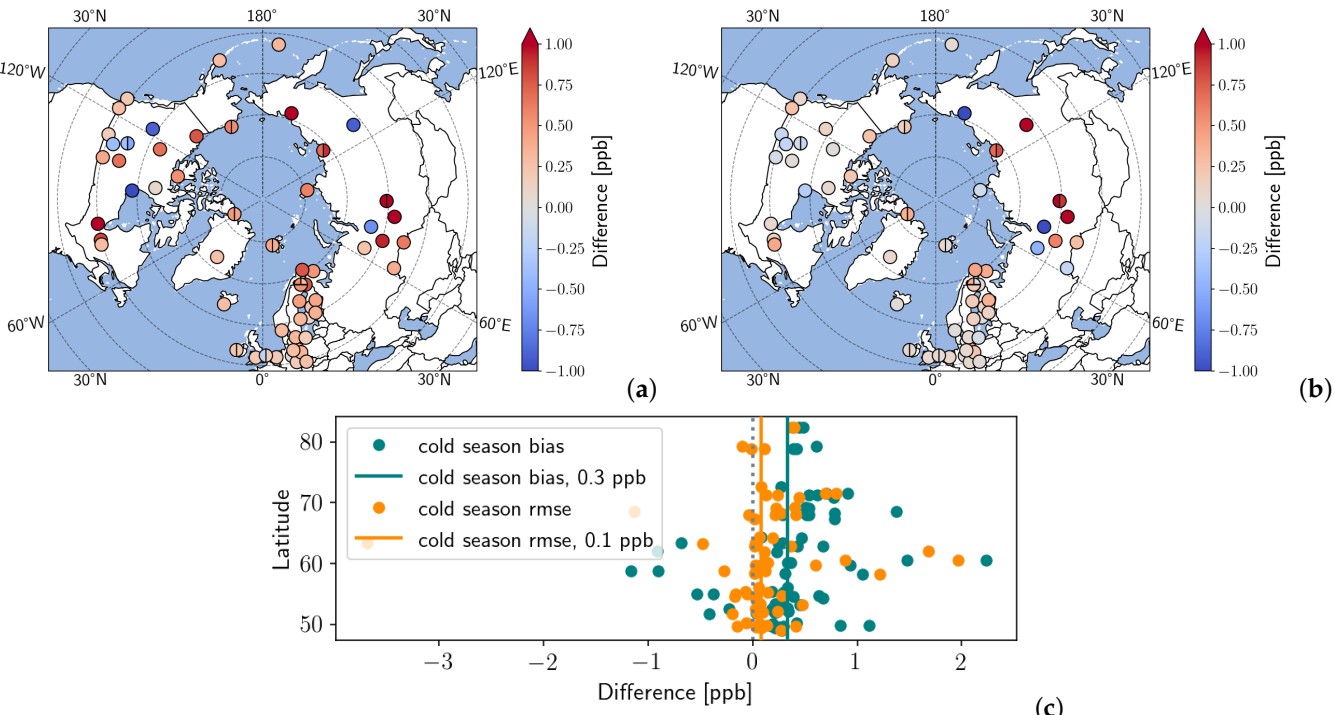

**Figure 3.** Differences in cold season biases (**a**) and in RMSEs (**b**) between mole fractions modelled with TM5 using LPX or LPX FT as the biospheric a priori. The statics are calculated against observations, i.e., model−observations. Positive values denote improved agreement with the observations, i.e., decrease in biases and RMSE, and negative values denote worsened agreement. (**c**) The cold season bias and RMSE differences between mole fractions modelled with TM5 using LPX or LPX FT as the biospheric a priori by latitudes. Median values are denoted with the solid vertical lines.

When the same comparison was done for the optimised mole fraction estimates without and with the SMOS F/T state (i.e., CTE−CTE FT), the median difference reduced to near zero for both the cold season and summer biases and the RMSEs. In Canada and Alaska, there were altogether 23 sites of which the cold season bias showed improved agreement with CTE FT compared to CTE in 65% sites and the cold season RMSE improved in 57% sites. In Europe, there were 31 sites in which also 65% showed improvement in cold season bias, but the cold season RMSE was improved only in 19% of the European sites. Russia had the remaining 13 sites, and there only 31% of sites showed improvement in cold season bias and 23% better cold season RMSE.

The sites where the changes in the statistics were among the lowest or highest 25% quantiles are shown in Figure 4 to highlight the most affected sites. The most affected sites did not show a clear pattern in their locations: both Eurasia and North America had sites with improved and worsened statistics and there were also no regions with only improved or worsened sites. This is most probably due to the nature of the atmospheric inversion, where it tries to resolve regional (and global) budgets by minimising the agreement to all observations.

We performed statistical significance tests on the median cold season and the summer bias and RMSE differences over the NHL sites. We used the Wilcoxon signed-rank test since the biases and RMSEs were not normally distributed. Both the bias and RMSE differences were statistically significant ($p < 0.05$) except the cold season and the summer RMSEs when the prior mole fractions were compared with each other.

The observations from the site in Baranova, Russia (79.28°N, 101.62°E), operated by the Finnish Meteorological Institute, were used for the first time in an inversion modelling study. The site's location is remarkably north (see Figure 4) and captures background Arctic air and, therefore, its data are noteworthy. The transport model was able to resolve the variations in the mole fractions well and the inversion was able to adjust the prior values well according to the site's data (see Supplementary Figure S4). Thus, the sites provide

background information about methane emissions in the Arctic, probably more for the Siberian side. The data may not provide additional information in the global inversion but could be useful to ascertain the emission estimates. It would also be valuable for validation and regional inversions. The implementation of SMOS F/T soil state did not have a substantial effect on the site's cold season and summer biases and RMSE values compared with the NHL sites. The cold season bias increased by 0.09 ppb and RMSE increased by 0.03 ppb. The summer values decreased a little: bias by 0.12 ppb and RMSE 0.03 ppb. Due to the site's location, the local $CH_4$ emissions are small in the a priori and thus, the SMOS F/T soil state implementation did not affect the site's mole fraction by much.

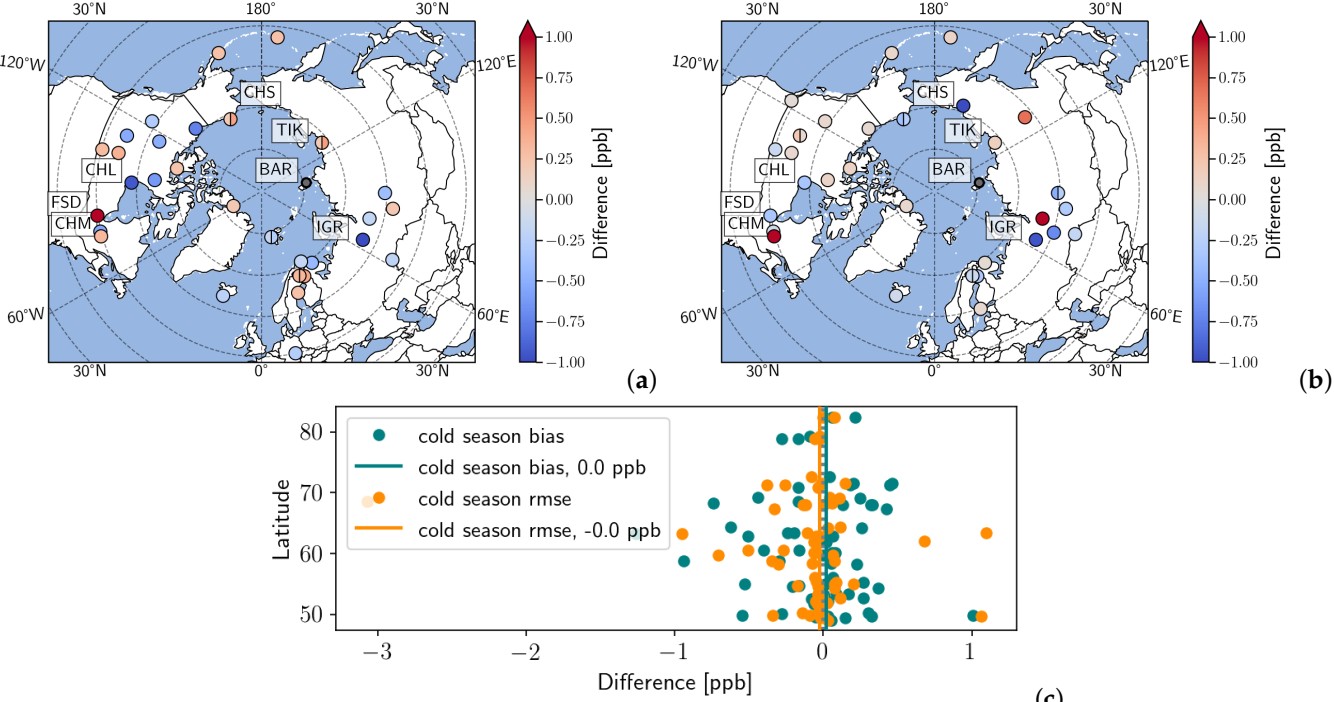

**Figure 4.** The cold season bias (**a**) and RMSE (**b**) differences between modelled mole fractions from CTE and CTE FT. The statics were calculated against observations, i.e., model−observations. Only values among the lowest or highest 25% quantiles are shown. Positive values mean that optimisation with the SMOS F/T soil state had smaller bias and RMSE than the optimisation without the soil state and, thus, had better agreement with the observations, and negative values denote worsened agreement. The sites with the largest differences in the cold season bias and RMSE (see Section 3.4) and Baranova's site (BAR) are labelled. (**c**) The cold season bias and RMSE differences between modelled mole fractions from CTE and CTE FT by latitudes. Median values are denoted with the solid vertical lines.

### 3.4. Sites with Large Differences in Cold Season Bias and RMSE

We selected six sites where the SMOS F/T soil status implementation affected the cold season biases and RMSEs the most for further inspection. The differences were calculated between CTE and CTE FT to examine the effect of the SMOS F/T implementation on the inversion. Three sites with the most improved and the most worsened values are presented. The cold season biases and RMSEs from mole fractions modelled with TM5 using LPX or LPX FT as a biospheric a priori as well as from optimised mole fractions from CTE and CTE FT are shown in Figure 5 for the selected six sites. The time series of observed and modelled mole fraction in each site is shown in Supplementary Figures S5 and S6. The six sites are also labelled in Figure 4. All sites have a sufficient amount of data that cover at least one winter and most of them have data covering almost the whole study period. Two of the most improved sites were in HBL and one on the northern coast of Russia. One of the most worsened sites was in HBL, one in WSL and one on the northern coast of Russia.

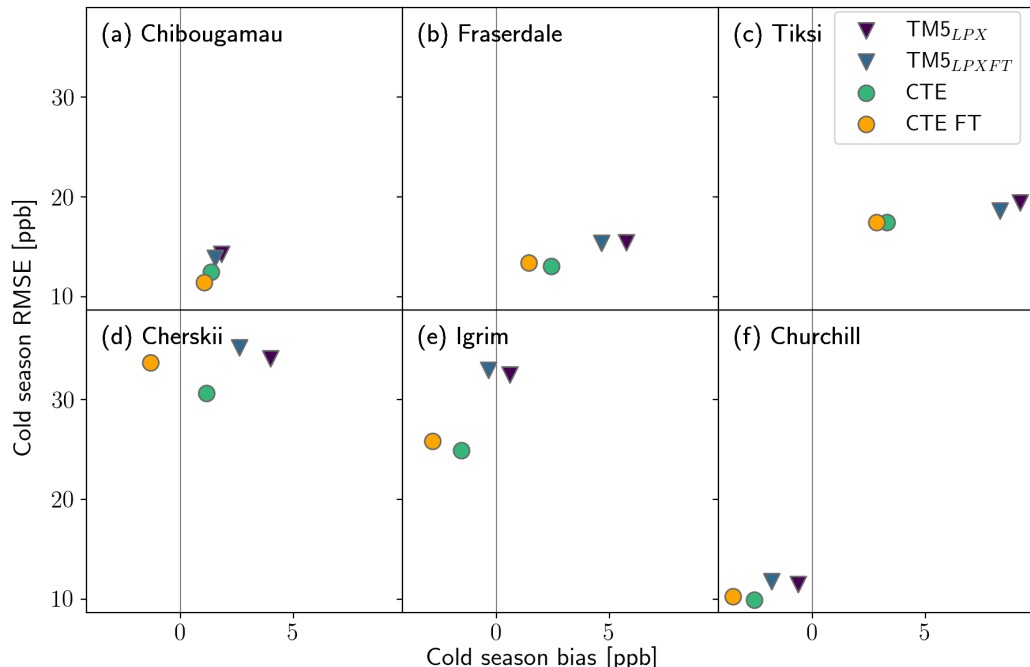

**Figure 5.** Detrended cold season bias and RMSE of modelled methane mole fractions compared with observation at selected sites. $TM5_{LPX}$ and $TM5_{LPXFT}$ are prior mole fraction modelled with TM5 using LPX or LPX FT as a biospheric a priori together with other sources. CTE and CTE FT correspond to optimised mole fractions from CTE-CH$_4$. (**a**) Chibougamau, Canada, (69.13°N, 105.06°W). (**b**) Fraserdale, Canada (49.88°N, 81.57°W). (**c**) Tiksi, Russia, (71.60°N, 128.89°E). (**d**) Cherskii, Russia (68.51°N, 161.53°E). (**e**) Igrim, Russia (63.19°N, 64.41°E). (**f**) Churchill, Canada (58.74°N, 93.82°W).

The site with the largest improvements in both the cold season bias and cold season RMSE was in Chibougamau (Canada, 69.13°N, 105.06°W). The difference in the cold season biases was 0.33 ppb and RMSE 1.06 ppb. The summer values, however, showed worsened agreement in Chibougamau: bias was increased by 3.94 ppb and RMSE by 2.52 ppb.

The cold season bias also improved in Fraserdale (Canada, 49.88°N, 81.57°W) by 1.06 ppb. Fraserdale is near Chibougamau. However, the cold season RMSE was increased by 0.34 ppb in Fraserdale. The summer values were decreased, i.e., improved.

In Tiksi (Russia, 71.60°N, 128.89°E), the mole fraction is measured with discrete (NOAA) and continuous (FMI) sampling methods. Comparison with continuous observations showed improvement of 0.46 ppb in the cold season bias but the cold season RMSE did not change much (increased by 0.01 ppb) whereas, in the discrete observations, the cold season bias was improved by 0.21 ppb and the cold season RMSE by 0.15 ppb. All the summer agreements were improved approximately as much as the cold season statistics. Altogether, the agreement with the observations improved in Tiksi.

The two sites where the agreement between the modelled and observed mole fractions decreased the most was in Cherskii (68.51°N, 161.53°E) and Igrim (63.19°N, 64.41°E) (Russia). In Cherskii, the cold season bias was increased by 0.17 ppb and the cold season RMSE by 3.07 ppb. In Igrim, the cold season bias was increased by 1.26 ppb and the cold season RMSE by 0.95 ppb. Interestingly, however, the summer bias and RMSE differences were improved (in Cherskii 3.29 ppb and 1.85 ppb and in Igrim, 1.76 ppb and 0.18 ppb).

The third site with the most worsened agreement between observations and modelled values was Churchill (Canada, 58.74°N, 93.82°W), which is also in HBL. There the cold season bias increased by 0.94 ppb and RMSE by 0.34 ppb. The summer values were also worsened by a similar amount by the same magnitude (the summer bias increased 0.90 ppb and the summer RMSE 0.13 ppb).

Almost all the selected six sites were in a region where permafrost is present [54]: Fraserdale, Igrim and Chibougamau were in regions with only isolated patches of per-

mafrost and Tiksi, Cherskii and Churchill in regions with continuous permafrost. During winter, the soil is frozen and methane emitted from the soil is minimal. Thus, the spikes seen in the observations are due to anthropogenic emissions transported to the site. As an example, we studied the winter spikes at Tiksi site in relation to wind speed and direction (see Supplementary Figure S7). In the northeast of the Tiksi site, there is the city of Tiksi at approximately 10 km from the site, from which anthropogenic methane signals could be transported to the observation site. The methane mole fraction spikes observed at Tiksi site occurred most commonly in summer, which are most probably caused by nearby wetland emissions. However, in the autumn of 2016 and the following winter, there are spikes when wind is blowing from the northeast, which could be caused by anthropogenic emissions. The winter spikes are not well captured in the modelled values, and the SMOS F/T soil implementation further reduces the magnitude of the modelled spikes. These sites are dominated by the biospheric emissions, and the anthropogenic emissions and their prior uncertainties around the sites are small, making it almost impossible for the inversion model to increase the anthropogenic emissions during such events.

The statistical significance of the differences between modelled optimised mole fractions without and with the SMOS F/T implementation was tested with the Wilcoxon signed-rank test for each selected six sites. The differences between cold season biases were statistical significance ($p < 0.05$) at all other selected sites except Tiksi. The differences between cold season RMSEs were statistically significant only at Chibougamau and at Cherskii.

### 3.5. The Impact of the SMOS F/T Implementation on Modelled Emissions

The SMOS F/T implementation affected the modelled atmospheric mole fraction but also methane emissions. Here, we concentrate on the NHL cold season and annual biospheric methane emissions of CTE FT and the difference between CTE FT and CTE (Table 2). The average cold season biospheric flux derived from CTE FT and its difference to CTE in the NHL are shown in Figure 6a,b. The largest differences in average cold season biospheric fluxes between CTE FT and CTE were found in the same regions where the SMOS F/T soil state implementation reduced the prior fluxes the most, i.e., in HBL and WSL.

**Table 2.** Annual and cold season average biospheric methane emissions of LPX FT and CTE FT, differences between CTE and CTE FT and the ratio of the difference to the CTE FT emissions. The difference LPX−LPX FT and the ratio of the difference to the LPX FT emissions are shown in brackets. The uncertainties are one standard deviation of ensemble distributions. Global emissions and emissions in the northern high latitudes (NHL), in Western Siberian Lowlands (WSL) and in Hudson Bay Lowlands (HBL).

| Annual Biospheric CH$_4$ Emissions | Global | NHL | WSL | HBL |
|---|---|---|---|---|
| LPX FT (Tg/year) | $123 \pm 41$ | $16.7 \pm 5.2$ | $4.2 \pm 3.3$ | $2.3 \pm 1.2$ |
| CTE FT (Tg/year) | $126 \pm 40$ | $23.1 \pm 4.0$ | $6.6 \pm 2.5$ | $2.8 \pm 1.0$ |
| Diff CTE−CTE FT (LPX−LPX FT) (Tg/year) | 0.75 (0.84) | 0.64 (0.80) | 0.19 (0.27) | 0.10 (0.12) |
| Ratio diff/CTE FT (diff/LPX FT) (%) | 0.60 (0.69) | 2.8 (4.8) | 2.8 (6.4) | 3.4 (5.3) |
| **Cold Season Biospheric CH$_4$ Emissions** | | NHL | WSL | HBL |
| LPX FT (Tg/season) | | $3.2 \pm 0.9$ | $0.73 \pm 0.5$ | $0.32 \pm 0.2$ |
| CTE FT (Tg/season) | | $3.5 \pm 0.8$ | $0.84 \pm 0.4$ | $0.27 \pm 0.2$ |
| Diff CTE−CTE FT (LPX−LPX FT) (Tg/season) | | 0.60 (0.72) | 0.20 (0.26) | 0.11 (0.13) |
| Ratio diff/CTE FT (diff/LPX FT) (%) | | 17.1 (22.6) | 23.2 (35.8) | 38.5 (40.8) |

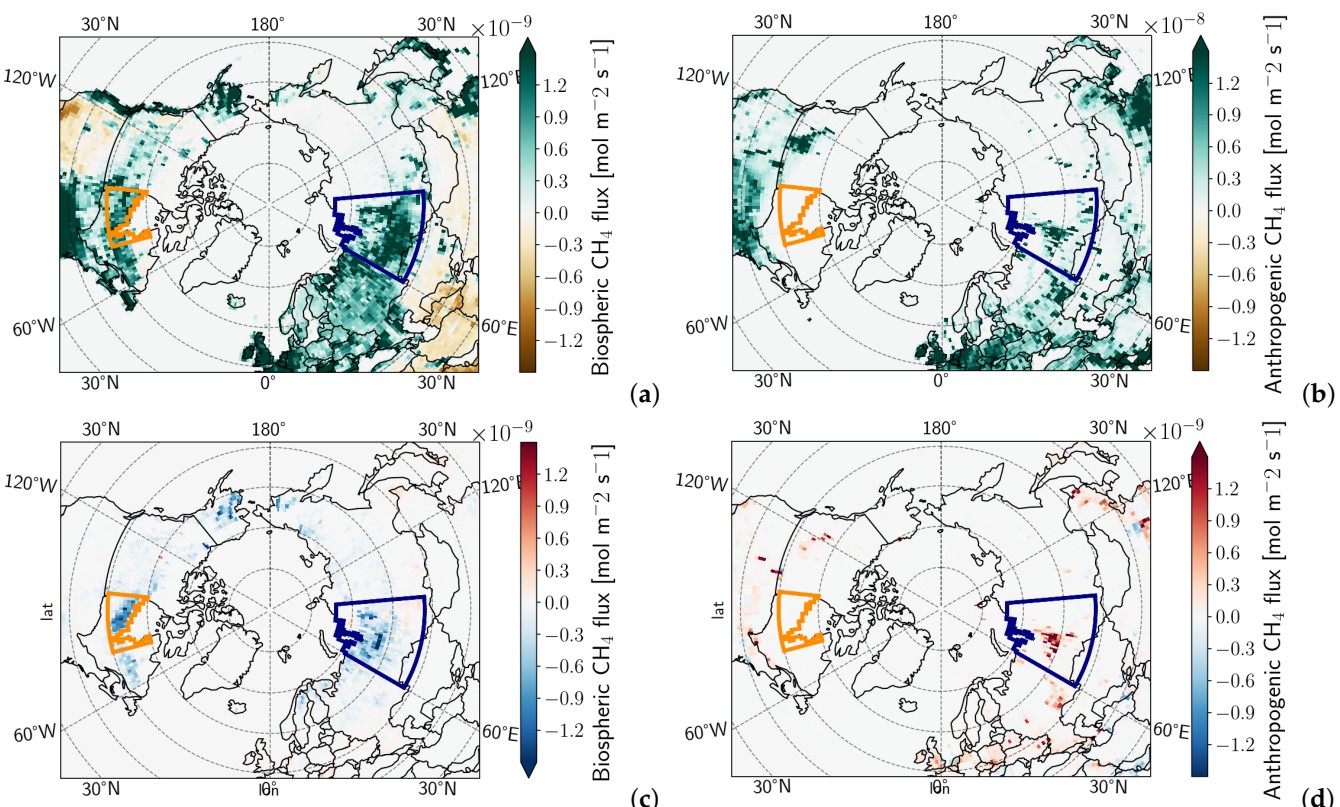

**Figure 6.** The average cold season (**a**) biospheric and (**b**) anthropogenic methane fluxes derived from CTE FT. The difference between the average cold season (**c**) biospheric and (**d**) anthropogenic methane flux of CTE FT and CTE. Note that the colour scale of anthropogenic cold season flux (**b**) is an order of magnitude larger than in the other figures. The circulated areas are Hudson Bay Lowlands (orange) and Western Siberian Lowlands (blue).

The annual and monthly biospheric emissions in the NHL are shown in Figure 7, in HBL in Figure 8 and in WSL in Figure 9. Figures include emissions from the priors, LPX and LPX FT, and the optimised emissions from CTE and CTE FT. The implementation of the SMOS F/T soil state hardly affected seasonal cycles on a regional scale. The average annual difference in the priors (LPX−LPX FT) was 0.84 Tg, but the difference between posterior emissions (CTE−CTE FT) was reduced to 0.75 Tg (Table 2). Similar changes, i.e., the difference between priors LPX and LPX FT was larger than the difference between posterior CTE and CTE FT, were also found in WSL and HBL, and during the cold season.

In HBL and WSL, the difference between monthly CTE and CTE FT was the largest in autumn (Figure 10a) but, especially in WSL, there was also a second drop in spring. The autumn drop was, however, much larger than the spring drop when the differences were compared to the absolute monthly emissions. The absolute monthly differences were larger in WSL but the relative changes were larger in HBL. As expected, the summer emissions (May–August) were not altered much and remained at the same level after the SMOS F/T soil state implementation.

Around the new year, the CTE FT was sometimes larger than CTE (for example, WSL in 2013 and HBL in 2015). This was mostly due to LPX FT being larger than LPX. The SMOS F/T implementation, thus, increased the mid-winter $CH_4$ emissions which was amplified in the optimisation leading to higher posterior emissions. The larger CTE FT emissions in WSL in 2013 were mostly responsible for the smaller change in the NHL scale.

We also studied anthropogenic emissions and emission differences, CTE FT−CTE, which are shown in Figure 6c,d. The reduced biospheric methane emissions in WSL were partly relocated to the anthropogenic emissions in the same region (Figure 6d). As visible in Figure 6c, the anthropogenic methane flux was larger than the biospheric flux in WSL in winter. The difference in the anthropogenic emissions between CTE FT and CTE was

inversely proportional to the difference in the biospheric emissions (Figure 10b). For every 1 Tg reduction in biospheric methane emission, WSL gained on average 0.36 Tg of anthropogenic methane emissions. Alternatively, in HBL where there were small anthropogenic emissions, there was no notable increase in the anthropogenic emissions. Thus, the excess biospheric methane had not been transferred to anthropogenic emissions and had been relocated elsewhere. The cold season and annual anthropogenic emissions defined with CTE FT and the difference between CTE FT and CTE in the NHL, in HBL and WSL are listed in Supplementary Table S2 with the global annual anthropogenic emissions and emission difference.

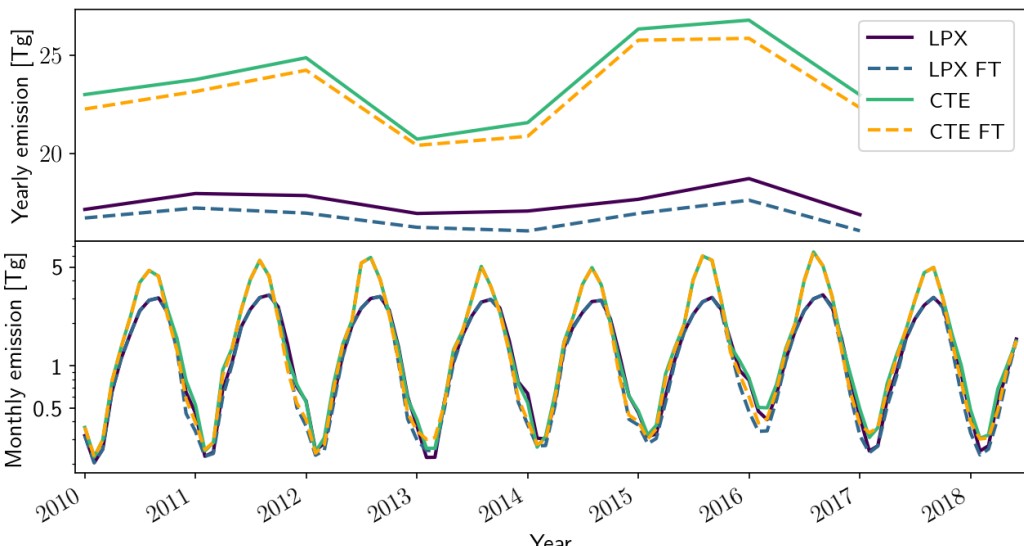

**Figure 7.** Yearly (**top**) and monthly (**bottom**) biospheric methane emissions in the NHL derived from the two priors, LPX and LPX FT, and two posteriors, CTE and CTE FT.

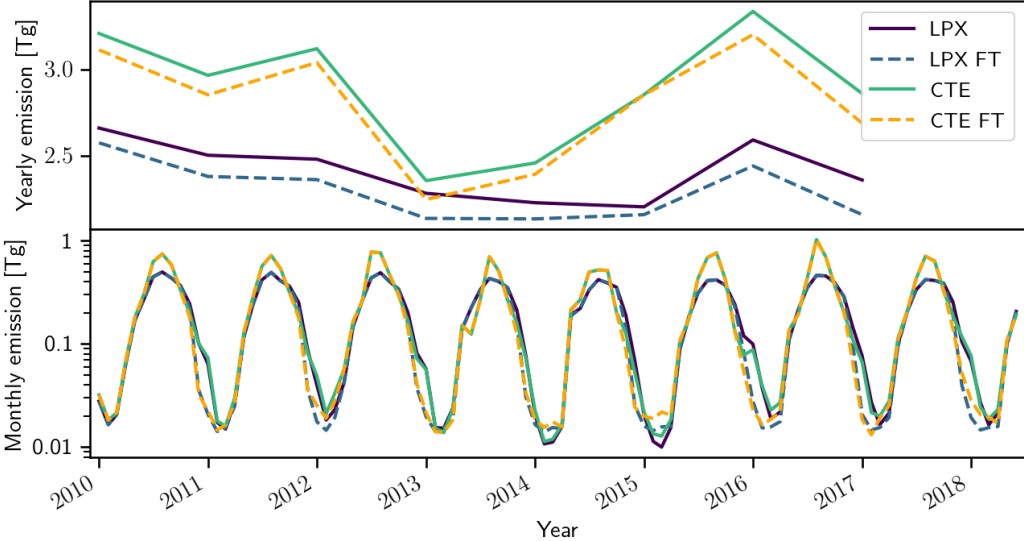

**Figure 8.** Yearly (**top**) and monthly (**bottom**) biospheric methane emissions in Hudson Bay Lowland derived from the two priors, LPX and LPX FT, and two posteriors, CTE and CTE FT.

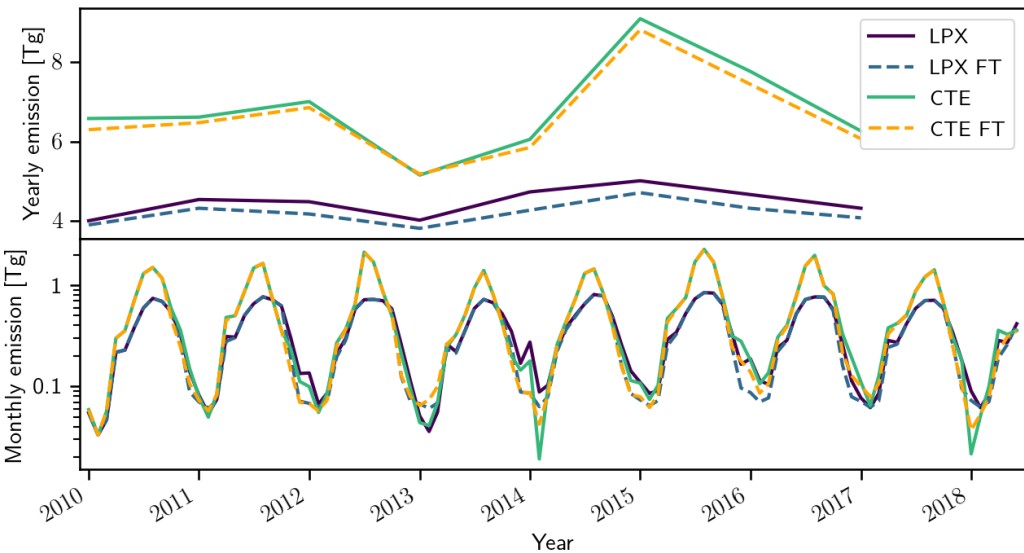

**Figure 9.** Yearly (**top**) and monthly (**bottom**) biospheric methane emissions in Western Siberian Lowlands derived from the two priors, LPX and LPX FT, and two posteriors, CTE and CTE FT.

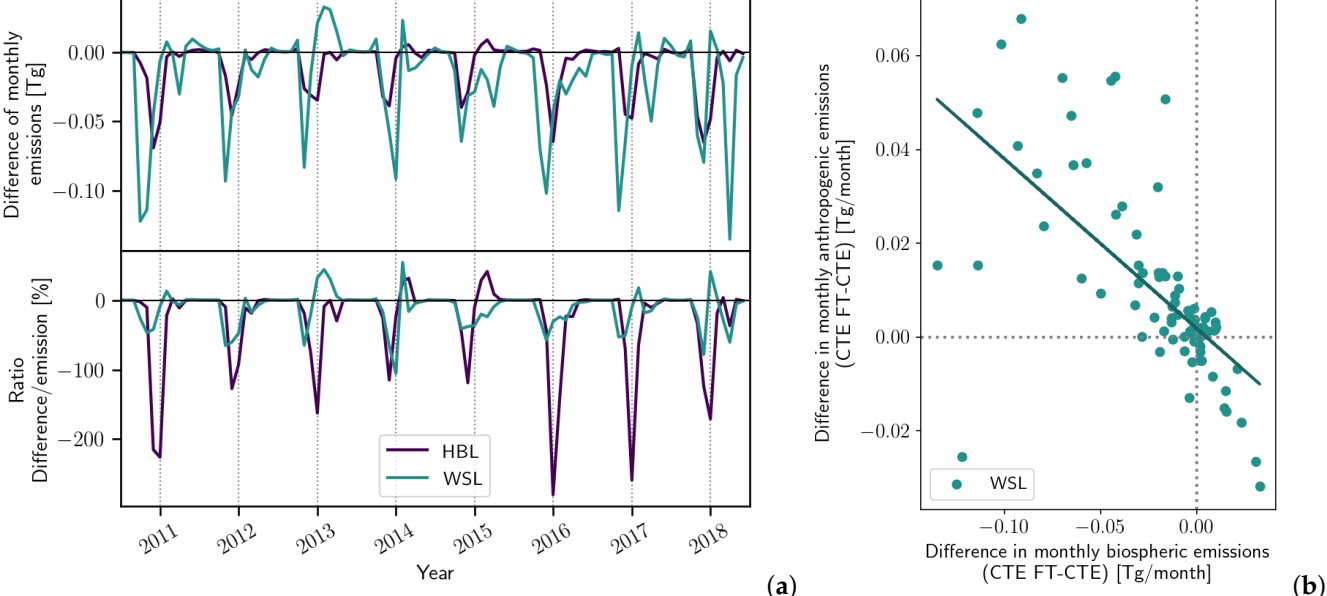

**Figure 10.** (**a**) Difference in monthly biospheric emission between CTE and CTE FT (top) and the ratio of the difference to the CTE monthly emissions (bottom). Values smaller than zero mean CTE is larger and larger than zero mean CTE FT is larger. Values are shown for both Hudson Bay Lowlands (HBL) and Western Siberian Lowlands (WSL) (**b**) Difference in monthly anthropogenic emission between CTE and CTE FT as a function of the difference in monthly biospheric emission in WSL between CTE and CTE FT. The ordinary least square linear regression is shown with solid lines (slope is $-0.36$ Tg/Tg and $R^2 = 0.44$).

We also conducted statistical significance tests on the differences between the optimised biospheric cold season methane emission estimates with and without the SMOS F/T implementation. The difference was statistically significant ($p < 0.05$) at an annual level only when the emissions were considered in the whole NHL. The differences in the cold season emission estimates were statistically significant ($p < 0.05$) also in WSL and HBL. None of the differences in the anthropogenic emissions was statistically significant.

## 4. Discussion

### 4.1. Uncertainties in the SMOS F/T Soil State and Its Implementation

Soil freeze/thaw state was used for the first time in an inversion model to constrain and estimate the cold season methane emissions. Thereby, the algorithm used here will probably go through further development. One could argue that the percentage of frozen grid cells which was used to define the wintertime was unnecessarily high (90% meaning that in large part, wintertime required all the SMOS F/T grid cells to be frozen) because it is probable that in autumn, methane flux reaches a wintertime level even before the soil is completely frozen since the emissions decrease from the summer maximum to the winter minimum gradually over weeks or even months [15,16]. Therefore, we could have also used more sophisticated implementation, for example, have a more subtle transition to wintertime fluxes.

The SMOS F/T algorithm is known to be less accurate in spring than in autumn because a wet snow layer attenuates the L-band signal significantly. Snow causes trouble, especially in spring because the snow accumulated during winter starts to melt from the top of the snow layer. The signal from wet snow resembles the signal from thaw soil, and thus, the algorithm is not as accurate to capture the springtime thawing as it is to represent soil freezing [24]. This feature of the SMOS F/T soil state algorithm in spring could be improved by adding an additional constraint with snow extent, which would also improve the SMOS F/T soil state implementation used here. Thus, even though the 90% frozen percentage is probably overcautious, as stated above, it is a highly confident indication of frozen soil given the flaws in the SMOS F/T algorithm.

One shortcoming of the SMOS F/T implementation is that it corrects LPX only when soil is frozen according to the SMOS F/T soil state but LPX-Bern DYPTOP is still emitting a notable amount of methane. Conversely, it does not correct LPX when soil is not frozen according to the SMOS F/T soil state, whereas there is no methane emitted by LPX-Bern DYPTOP. This shortcoming introduces a low bias in the methane emission when using the SMOS F/T implementation. Due to uncertainties related to spring, the current version of the SMOS F/T soil state would not be suitable to define soil thawing in spring, but it is accurate enough that it could be used to correct too early soil freezing in LPX-Bern DYPTOP. However, defining autumn methane emissions is not as straightforward as defining the cold season emissions. Rather than using the SMOS F/T soil state to postprocess LPX-Bern DYPTOP's emissions estimates, it would be better to use it already within LPX-Bern DYPTOP when the autumn methane emissions are modelled.

Another problem with the SMOS F/T algorithm concerns the operation frequency of SMOS. SMOS operates at L-band reserved for radio astronomy and it is supposedly free of any human-made radiation as agreed by the International Telecommunications Union. However, there are regions heavily polluted by radio frequency interference (RFI) in the SMOS L-band. The most problematic areas are over China and the Middle East where there are spots with totally polluted areas [55] but the RFI also limits the number of SMOS observations in Eurasia, especially in the southern and western border of Russia. There have been efforts to minimize the effect of the RFI, and when the next version of SMOS data is released, the SMOS F/T soil state will also be updated. The update would improve our defined wintertime periods and the SMOS F/T implementation.

The RFI detection and filtering in SMOS data are done in several levels and with multiple algorithms [55]. The detection and filtering are highly effective and, thus, the data, which SMOS F/T soil state uses, have no RFI polluted data points to a significant extent. The RFI and its filtering are shown as missing data points which, on the other hand, do not alter methane emissions in our SMOS F/T implementation. Missing data points do affect the SMOS F/T soil state to some extent since there are fewer data in general for averaging but the SMOS F/T algorithm has its own quality flagging system to remove such uncertain pixels. Thus, the data gaps had led mainly to shorter wintertime periods meaning the SMOS F/T soil state implementation had likely a smaller impact than it would have without the data gaps.

*4.2. The Start and End of the Cold Season*

Modifying the biospheric methane emissions based on the SMOS F/T soil state affected emissions more in autumn when the cold season started than in spring when the cold season ended. This is mainly due to the clear differences at the start and the end of the cold season defined by the SMOS F/T soil state and LPX-Bern DYPTOP (Figure 2). At the start of the cold season, the methane emissions are affected only after the SMOS F/T soil state freezes, e.g., the main effect is in regions and years when the SMOS F/T freezes earlier than LPX-Bern DYPTOP. This is especially observed in HBL and in WSL during almost all years. In spring, at the end of the cold season, the SMOS F/T shows soil thawing even months before LPX-Bern DYPTOP, meaning the SMOS F/T implementation does not affect the spring emissions by much.

The reason why the SMOS F/T soil state shows earlier soil freezing than LPX-Bern DYPTOP could be explained by the snow scheme in LPX-Bern DYPTOP. Snow and its insulation are hard but important to represent in the ecosystem models [20,21]. Snow insulation is important when modelling permafrost's active layer thickness but also when modelling the freeze/thaw cycle of the soil in non-permafrost regions [20]. Effective snow depths calculated by Burke et al. (2020) [19] showed that ecosystem models used in the CMIP6 models tend to overestimate the effective snow depth in the tundra. The snow also extended further south compared to observations. Furthermore, ecosystem models represent the topsoil temperatures better during a snow-free period than when there is snow present [20]. In LPX-Bern DYPTOP, the snow could arrive too early insulating the soil and lengthening the freezing period or the described snow insulation could be too strong. The snow is modelled in LPX-Bern DYPTOP as precipitation, which falls when the daily air temperature is lower than −2 °C. Furthermore, the representation of insulation of vegetation cover could have flaws [20]. The depth of the modelled soil in LPX-Bern DYPTOP is relatively shallow: only the top 2 m is modelled with detailed soil layers and 8 m padding beneath it is modelled as one layer [56] which is shown to be too shallow, especially in permafrost regions [19]. However, the differences between the start of the cold season rarely exceed a month, meaning the time resolution in LPX-Bern DYPTOP could be the main reason why the SMOS F/T soil state shows earlier soil freezing.

The reason why the SMOS F/T soil state implementation less affected the end of the cold season is probably mainly due to the characteristics of the SMOS F/T algorithm. Because the detection of the soil freeze/thaw state is based on whether the detected water is liquid or ice, wet snow can be classified as thawed soil even though the soil beneath can still be frozen, as discussed above. This also decreases the reliability of our analysis regarding the spring emissions.

With the current version of the SMOS F/T soil state, the modifications done to biospheric methane emissions can be thought of as reliable, especially during the start of the cold season. The SMOS F/T soil state is less reliable at the end of the cold season but in a way that did not affect the original biospheric methane emissions. Thus, we cannot say that the spring emissions were corrected to represent the truth but it also did not change the spring emissions to be more unreliable.

*4.3. Atmospheric Inversion and Methane Mole Fractions Estimates*

Modelled and optimised $CH_4$ atmospheric mole fractions were compared to observations to study whether the modelled atmospheric mole fractions matched better with the observations when the SMOS F/T soil state was used in the optimisation. The observations were also assimilated in the optimisation (excluding those which were left out because their quality was not sufficient or they differed too much from the modelled mole fraction (see Section 2.3)), so it is expected that the posterior mole fractions are closer to the observations than the prior mole fractions (as seen in Supplementary Figure S3). It should also be noted that comparing assimilated data to model estimates is only one way to quantify the model performance, and comparison to independent observation is recommended since the model naturally modifies the estimates to match the observations. However, we will likely get the

best modelling results by using all available and reliable data in the optimisation. Testing the model performance against assimilated observations is also a conventional practice among inversion models, e.g., [44,57,58].

The posterior atmospheric mole fractions estimated with the modified biospheric a priori in which the soil state was implemented did not show universally better or worse agreement with the observations than the optimised mole fractions with the original LPX as the biospheric a priori. In theory, the inversion should end up in the same concentration estimates despite the priors. Thus, it is also a sign that our data assimilation system is working when we see similar concentration estimates with different priors. In practice, we do not have enough observations, and assimilated observations are not exactly the same, and therefore, we expect the model estimates to be affected by the different priors also when they differ only partly from each other (here, only during wintertime and only in the NHL). Additionally, the degrees of freedom in the flux multipliers for biospheric and anthropogenic sources were different, and a priori fluxes for those sources were assumed to be independent, which allowed the inversion to optimise those separately, ending up with different total flux distributions.

The mole fractions modelled with TM5 with the two biospheric priors showed improved agreement with the observations when the SMOS F/T soil state was used (Figure 3). This indicates that the modified biospheric a priori with the soil state estimates implemented was more realistic than the original a priori. However, the improvement in priors' mole fractions did not directly translate into optimised mole fractions in all sites. We suspect that this could be explained by too small anthropogenic a priori and the way we define the uncertainties in the data assimilation: CTE-$CH_4$ might want to incorrectly increase biospheric emissions in the situation where anthropogenic emissions should be increased but their a priori emissions are too small and, thus, also their given uncertainties are small (the uncertainty of the prior fluxes over land is assumed to be 80%). If the total a priori methane emissions are too small, but the model cannot increase anthropogenic emissions, the only possibility is to increase biospheric emissions. Thus, too small given uncertainty could prevent the model from increasing the emission of the correct source and the model locates excess methane for the other source (if the other source's uncertainty is larger). This could be prevented, for example, by defining the uncertainty of a priori flux as some absolute value instead of a relative value.

### 4.4. Cold Season Methane Emissions

The cold season biospheric emissions were decreased mainly in HBL and WSL. The decreased biospheric emissions were partly transferred to anthropogenic emissions, especially in WSL (Figures 6 and 10b). This is an inversion feature trying to reserve the total fluxes to match atmospheric observations of methane. WSL has both large biospheric and anthropogenic methane sources [5,50], which complicates the distinction of the magnitude of each source in the area. Thus, it is important that a priori fluxes are as reliable as possible in the region.

Thompson et al. (2017) [58] estimated the monthly methane emissions in northern latitudes with an inversion model, which used the Lagrangian particle dispersion model FLEXPART [59]. They estimated the annual anthropogenic methane emissions to be even higher in WSL (12.7 ± 3.6 Tg/year) than estimated here (7.3 ± 1.9 Tg/year), while the biospheric emission estimates were similar to our analysis (6.9 ± 3.6 Tg/year and our estimation: 6.6 ± 2.5 Tg/year). Likewise, Wang et al. (2019) [60] also showed that emissions in Russia may have been underestimated in EDGAR v4.3.2 compared to those reported to the UNFCCC and inferred from the GOSAT observations.

Peltola et al. (2019) [14] used random forest machine learning to upscale methane eddy covariance flux measurements in the northern latitudes. The annual biospheric emissions in HBL range from 2.3 Tg/year to 5.4 Tg/year according to Peltola et al. (2019) and references within [14]. Our estimate was 2.8 ± 1.0 Tg/year, and the impact of SMOS F/T soil state was 0.10 Tg/year which puts our estimate at the lower limit of the range of the earlier

estimates. The annual emission range in WSL, on the other hand, was 4.2–7.0 Tg/year [14]. Here, we estimated the annual biospheric $CH_4$ emission to be $6.6 \pm 2.5$ Tg/year and the impact of SMOS F/T soil state was 0.19 Tg/year. This is closer to the upper limit.

Peltola et al. (2019) [14] also estimated non-growing season (November–March) wetland emissions in northern latitudes above 45°N to be 4.5–8 Tg/season. Treat et al. (2018) [13] synthesized non-growing season methane emissions above 40°N and estimated emissions to be $6.1 \pm 1.5$ Tg/season. Our estimates were a little lower than these, $3.5 \pm 0.8$ Tg/season, though, we considered emissions only above 50°N, and our study period was until the end of April. Thus, our cold season emissions are at the lower limit of the earlier estimates and even more so after the SMOS F/T soil state implementation.

As observed, the ranges in the cold season biospheric methane emission and the year-to-year variability are large (here, 3.0–4.2 Tg/cold season). When we compare that to the effect of implementing the SMOS F/T soil state to CTE-CH$_4$ (0.60 Tg/season on average), the effect is too small to strongly conclude the confidence in one direction or another. Furthermore, similar to the mole fractions, the optimisation had a larger effect than the SMOS F/T implementation, i.e., the difference between the posterior and the prior (CTE FT−LPX FT) was larger than the difference between the two posteriors (CTE FT−CTE). In other words, the data assimilation with the mole fraction observations did the "heavy lifting" work and the SMOS F/T implementation fine-tuned the emission estimates. Nevertheless, even though the change in the cold season emission was small compared to the annual emission, the relative change was still large (17–38% depending on the scale considered) since the cold season emissions are small. Furthermore, the fine-tuning did not stop on our main target, the biospheric emissions, but it extended over the anthropogenic emissions and implied a higher significance of anthropogenic emissions in the cold season. However, it is not straightforward to find independent evidence that distribution between biospheric and anthropogenic emissions was improved, especially when the changes were relatively small. Nevertheless, since the inversion model is at its best when estimating the total emissions, this means that when we have better biospheric emission estimates, we also automatically estimate the anthropogenic part of the emissions better.

Considering the points above, we can conclude that the estimated methane emissions with the modified biospheric a priori are more realistic. It also indicates that the cold season anthropogenic emissions in the NHL need to be better quantified. We expect that our analysis could be improved if we changed the way we define the uncertainty of different emission categories in the data assimilation. This would benefit especially the anthropogenic emission estimates and highlight their importance in the cold season.

## 5. Conclusions

We developed a novel method to use remote sensing soil freeze/thaw status in improving the seasonality of cold season methane emissions in the northern high latitudes. Daily soil freeze/thaw state values from the SMOS F/T soil state were used to define more accurately the timing of soil freezing and thawing in the NHL, and biospheric methane emission estimates from LPX-Bern DYPTOP was modified according to defined cold season periods. LPX-Bern DYPTOP's advantage is its spatial accuracy, but it lacks validation of freezing dynamics, which motivated our SMOS F/T implementation.

The original methane emission estimates from LPX-Bern DYPTOP (LPX) and its modified version (LPX FT) were used as a biospheric a priori in an atmospheric inversion model CTE-CH$_4$, and the differences between modelled cold season methane emissions and mole fractions were compared. The effect of SMOS F/T implementation on the cold season biospheric methane emissions was 0.6 Tg after the optimisation. The largest impact was in WSL and in HBL, where large wetland areas are located. Furthermore, anthropogenic emission increased on average by 0.23 Tg/cold season in the same area, compensating part of the decrease in the biospheric emissions.

Before the optimisation, modelled mole fractions agreed better with observations when the SMOS F/T soil state was used, but after the optimisation, the agreement with

observations was not greatly improved any more. The decrease in the cold season methane emissions, although statistically significant, was relatively small and, thus, might not be seen undoubtedly from modelled mole fractions. Furthermore, the way uncertainties were defined in our data assimilation system (related to the magnitude of the emissions) might have prevented the optimisation from adjusting biospheric and anthropogenic emissions to some extent. To better separate biospheric and anthropogenic emissions in regions where they both are important sources, we could, for example, use additional information from other tracers such as isotopes and co-emitted species (CO, ethane) in the inversion.

Regardless, we could show that using the SMOS F/T soil state to define the cold season period was an advantageous method to define the cold season methane emission in the NHL. The SMOS F/T soil state is also a potential data source to study other aspects of the seasonality of northern wetlands methane emissions, such as soil freezing emissions. It is also recommended to continue developing methods, which combine atmospheric inversion models and different remote sensing data sources.

**Supplementary Materials:** The following are available online at https://www.mdpi.com/article/10.3390/rs13245059/s1, Table S1: List of surface observation sites used in inversions, located below 49°N. Observation Uncertainty (Obs. Unc.) is used to define diagonal values in the observation covariance matrix. Data type is categorized into two measurements (discrete (D) and continuous (C)); Table S2: Annual and cold season average biospheric methane emissions of EDGAR v5.0 and CTE FT, differences between CTE and CTE FT and the ratio of the difference to the CTE FT emissions. The uncertainties are one standard deviation of ensemble distributions. Global emissions and emissions in the northern high latitudes (NHL), in Western Siberian Lowlands (WSL) and Hudson Bay Lowlands (HBL) are shown; Figure S1: Daily averaged $CH_4$ mole fraction observations [ppb] in measurement stations in years 2010–2018. Stations are ordered by their latitudes (the northernmost at the top and the southernmost at the bottom, only stations above 49°N are shown); Figure S2: Figure illustrates the optimisation regions, (a) anthropogenic and (b) biospheric sources. We applied grid-based optimisation in the NH Temperate to Arctic regions, where observation network is the most dense. For other areas, region-wise optimisation is used, where the regions are defined based on modified TransCom regions and soil-type distribution from LPX-Bern DYPTOP model. The grid-based optimisation is applied to the NH Temperate to Arctic regions. $1° \times 1°$ latitude $\times$ longitude grid-based optimisation is applied to the wetland regions (R1), surrounded by $2° \times 3°$ intermediate grids (R2). Elsewhere we have $4° \times 6°$ optimisation resolution (R3). We applied correlation lengths of 100, 200 and 500 km, for R1, R2 and R3, respectively. The correlation length for region-wise optimisation regions is 500 km over land and 900 km over ocean; Figure S3: Differences in cold season biases (a) and in RMSEs (b) between mole fractions modelled with TM5 using LPX as a biospheric a prior and optimised mole fractions from CTE FT. The statics are calculated against observations, i.e. model−observations. Positive values denote improvement in agreement with the observations, i.e. decreases in biases and RMSE, and negative values denote worsened agreement. c) The cold season bias and RMSE differences between mole fractions modelled with TM5 using LPX as a biospheric a prior and optimised mole fractions from CTE FT by latitudes. Median values are denoted with the solid vertical lines; Figure S4: Observed and modelled CH4 mole fractions at Baranova, Russia (79.28°N, 101.62°E). Modelled mole fractions without optimisation (LPX, LPX FT as the biospheric a priori) and with optimisation (CTE, CTE FT) are shown; Figure S5: Observed and modelled CH4 mole fractions at selected measurement sites. Observations are daily averages used in the inversion (Section 2.3). Modelled mole fractions without optimisation (LPX, LPX FT as the biospheric a priori) and with optimisation (CTE, CTE FT) are shown. (a) Chibougamau, Canada, (69.13°N, 105.06°W). (b) Fraserdale, Canada (49.88°N, 81.57°W). (c) Tiksi, Russia, (71.60°N, 128.89°E); Figure S6: Observed and modelled CH4 mole fractions at selected measurement sites. Observations are daily averages used in the inversion (Section 2.3). Modelled mole fractions without optimisation (LPX, LPX FT as the biospheric a priori) and with optimisation (CTE, CTE FT) are shown. (a) Cherskii, Russia (68.51°N, 161.53°E). (b) Igrim, Russia (63.19°N, 64.41°E). (c) Churchill, Canada (58.74°N, 93.82°W); Figure S7: (a) Wind speed and methane concentration at Tiksi site. The observations are hourly means from 12 pm to 16 pm and filtered based on a quality flag. Northeastern (0°–90°) wind, i.e., wind blowing from the direction of a nearby city, is highlighted with green. (b) Wind speed and direction distribution at Tiksi site.

**Author Contributions:** M.T., A.T. and T.A. participated on the conceptualization of the study; M.T. did the data processing, prepared and performed the model runs and did the visualizations for the manuscript; M.T. did the analysis and prepared the original manuscript together with A.T., T.A. and K.R.; A.T. helped with setting up the CTE-CH$_4$ and TM5 model runs; K.R. provided the SMOS F/T soil state data and helped to analyse the results involving the soil state; T.A. advised the analysis and writing process in general; V.K. participated in defining the cold season limits in the data analysis; R.E. provided the atmospheric observations from the Giordan Lighthouse station. All authors have read and agreed to the published version of the manuscript.

**Funding:** This research was funded by the Academy of Finland (307331 UPFORMET), European Space Agency (ESRIN Contract No: 4000125046/18/I-NB ESA-MethEO), and European Union (776810 EU-H2020 VERIFY). The SMOS F/T product was funded by Academy of Finland (309125 CRYO-TREC), and European Space Agency (ESRIN Contract No: 4000124500/18/I-EF SMOS F/T Service).

**Institutional Review Board Statement:** Not applicable.

**Informed Consent Statement:** Not applicable.

**Data Availability Statement:** The model results and code will be provided on request from the corresponding author (Maria Tenkanen, Maria.Tenkanen@fmi.fi). The SMOS F/T soil state can be downloaded from ESA SMOS dissemination server [31] and from FMI [32].

**Acknowledgments:** We thank the team behind the LPX-Bern DYPTOP v1.4 for providing the model estimates, and Jurek Müller and Sebastian Lienert for their assistance in interpreting the results with LPX. We are grateful for Agencia Estatal de Meteorología (AEMET), CSIRO Oceans and Atmosphere, Dirección Meteorológica de Chile (DMC), Environment and Climate Change Canada (ECCC), the Environmental and Chemical Processes Laboratory, University of Crete (ECPL/UOC), the Hungarian Meteorological Service (HMS), the Institute for Atmospheric Sciences and Climate (ISAC), Laboratoire des Sciences du Climat et de l'Environnement (LSCE), the National Institute of Water and Atmospheric Research Ltd. (NIWA), the Environment Division Global Environment and Marine Department Japan Meteorological Agency (JMA), the Main Geophysical Observatory (MGO), Meteorology, Climatology, and Geophysics Agency Indonesia (BMKG), the Max Planck Institute for Biogeochemistry, National Institute for Environmental Studies (NIES), Netherlands Organisation for Applied Scientific Research (TNO), Norwegian Institute for Air Research (NILU), Ricerca sul Sistema Energetico (RSE), the Swiss Federal Laboratories for Materials Science and Technology (EMPA), Umweltbundesamt Germany/Federal Environmental Agency (UBA), Umweltbundesamt Austria/Environment Agency Austria (EAA) as the data provider for Sonnblick, the Southern African Weather Service (SAWS), University of Bristol (UNIVBRIS), University of Exeter, University of Urbino (UNIURB), and University of Wisconsin-Madison (UofWI) for performing high-quality CH$_4$ measurements at global sites and making them available through the GAW-WDCGG and personal communications. In situ observations collected over the US Southern Great Plains were supported by the Office of Biological and Environmental Research of the US Department of Energy under contract no. DE-AC02-05CH11231 as part of the Atmospheric Radiation Measurement (ARM) Program, ARM Aerial Facility (AAF), and Terrestrial Ecosystem Science (TES) Program.

**Conflicts of Interest:** The authors declare no conflict of interest. Furthermore, the funders had no role in the design of the study; in the collection, analyses, or interpretation of data; in the writing of the manuscript, or in the decision to publish the results.

## Abbreviations

The following abbreviations are used in this manuscript:

| | |
|---|---|
| NHL | Northern High Latitudes |
| WSL | Western Siberian Lowlands (52–74°N, 60–94.5°E) |
| HBL | Hudson Bay Lowlands (50–60°N, 75–96°W) |
| SMOS | Soil Moisture and Ocean Salinity |
| SMOS F/T | SMOS Freeze/Thaw soil state |
| CTE-CH$_4$ | CarbonTracker Europe-CH$_4$ |

| LPX | Methane emission estimates of LPX Bern DYPTOP v1.4 (a dynamic global vegetation and land surface process model) |
| LPX FT | LPX with the SMOS F/T implementation done in this study |
| CTE | Methane emission (and concentration) estimates optimized with CTE-$CH_4$ and LPX as the biospheric a priori |
| CTE FT | Methane emission (and concentration) estimates optimized with CTE-$CH_4$ and LPX FT as the biospheric a priori |

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
