# Peer review of "Utilizing Earth Observations of Soil Freeze/Thaw Data and Atmospheric Concentrations to Estimate Cold Season Methane Emissions in the Northern High Latitudes"

_remotesensing, doi:10.3390/rs13245059_

Round 1

Reviewer 1 Report

This manuscript presents a new method to improve the representation of frozen soils in the LPX model based on measurements from the SMOS satellite instrument, with the aim to improve estimates of methane emissions at high northern latitudes. The modified methane emissions are evaluated using measurements of the atmospheric mixing ratio of methane at northern sites, in combination with the Carbon Tracker Europe inversion system. Whereas the a priori mixing ratios show an improvement using the new SMOS F/T data, it is less clear that the inversion-optimized fluxes benefit from the updated a priori fluxes. Suggestions are made to further improve the use of the SMOS F/T data to improve the LPX estimates, as well as the general setup of the inversion.

Overall, the manuscript makes a useful contribution in my opinion. Some parts need a careful read by a native English speaker. Somewhere I ended up editing sentence by sentence, which I decided to give up on since I do not consider this as my job. As explained further below, also some scientific issues need further attention to make this manuscript suitable for publication. However, nothing outside the range that should be possible to repair.           

GENERAL COMMENTS

Many differences are shown in discussed for which it is unclear how significant they are. There is little mentioning of estimation uncertainties and how differences compare with those uncertainties. It may be that a considerable amount of text is spent on differences that essentially do not mean anything in a statistical sense. This is true for fluxes as well as simulated mixing ratios (which may be somewhat more difficult to address).

For what is called LPX and LPX FT it is often not clear if it concerns only the biospheric contribution to the a priori flux or the total a priori flux. Sometimes LPX and LPX FT refer to CTE simulated mixing ratios using those fluxes, which also works confusing. The names that are used should be reconsidered to avoid such confusions.

One shortcoming of the modified biospheric fluxes as that they only correct LPX when it misses a frozen soil. Conversely, however, it does not seem to correct LPX when the soil is not frozen whereas it is according to LPX. This method obviously introduces a low bias in the methane emission, which currently is not mentioned or discussed in the context of the results.

Another shortcoming of the proposed algorithm is that methane emissions are assumed to stop when the soil freezes. This has been discussed in the literature, in the context of measurements indicating that emissions may still occur in the presence of vegetation that is rooted deep enough to reach still unfrozen layers. Interestingly, the inversion appears not to undo F/T-introduced emission reductions in the fall season. On the other hand, the posterior solution seems to follow the prior on this difference, suggesting that there is simply not enough atmospheric information to resolve it. More discussion on this would benefit the significance of the scientific message of this work.             

It is suggested that the use F/T information helps improve the attribution between high northern biospheric and anthropogenic emissions. However, independent evidence is missing that the shifts in emission attribution that are obtained are in fact improvements. Some discussion about this should be added.    

SPECIFIC COMMENTS

Line 57, SMOS is the first mission using a passive sensor in the L-band. The word passive needs to be added, since radars using the L-band have been in orbit longer ago (e.g. JERS / ALOS PALSAR). 

Line 127, The description of how uncertainties were defined is very qualitative. This makes it difficult to understand / verify the numbers in Table 1 and S1. Some more guidance is needed.

Line 133: ‘..did not affect much …’ affect what? (number of measurements? Inversion optimized fluxes?...)

Line 137: Emissions from termites are also biospheric emissions. Because of this it is unclear what is meant with “biospheric”.

Line 147: What ECMWF data provide oceanic methane sources? Reference 28 does not refer to an ECMWF related document.

Line 154: ‘Note …’ Does this mean that fluxes were estimated at a higher resolution than the resolution of the transport model? If so, what information gain do you expect from this?

Line 160: Why would the correlation length if the a priori flux uncertainty vary with observational density?

Line 162: I suppose that rather than no temporal correlation, you assume a temporal correlation of zero? What justifies this choice?

Line 179: ratio between what and what?

Line 182: So, then what do you do with the biospheric fluxes for the last year?

Line 183: Given the potential importance of non-growing season CH4 fluxes, what motivates the use of the lowest emission (which supposedly is close to zero) under freezing conditions?

Line 212: ‘around the New Year’ I would have expected the coldest day of the year to be later than New Year, because of the thermal inertia in the climate system.

Line 227: ‘..change..’ between what and what?

Line 239: Why is the resolution of the time axis in Figure 2 daily, whereas the time resolution of LPX is monthly? Some fraction of the differences that are shown should just be caused by a difference in temporal resolution. This caveat of Figure 2 should be made clearer.  

Line 302, ‘..its data is noteworthy’ why is this comparison not shown?

Line 313, How significant are the differences that are shown here compared with the measurement and a posteriori mixing ratio uncertainties?  

Line 351, Dominated by biospheric or anthropogenic emissions?

Line 376, How can LPX FT have larger fluxes than LPX?

Line 392, The manuscript ends with a discussion section. What I am missing is a conclusion section. An option would be to call section 4 ‘Discussion and conclusions’, but given the length of the section it would be better to have a separate section summarizing the main conclusions of this work.   

Line 397, ‘cold season’ instead of ‘wintertime’?

Line 403, What is meant by ‘weaker’ here? Less accurate?

Line 417, ‘some regions are totally polluted’ Which are they, and how important could they be?   

Line 419, ’updated version of SMOS is coming’ What is the source of this information?

Line 420-421: RF pollution would cause an overestimated impact of the FT algorithm if it remains undetected by the SMOS filtering algorithm. How do we know which of the two options has the largest impact?

Line 429-430: ‘the SMOS FT … by much’ Doesn’t this mean that in LPX the emissions should start earlier in the season? Is there a way to inform LPX about the SMOS observed FT state?

Line 443: ‘The snow is modelled in LPX …’ Do you think that this could explain why snow may fall too early in LPX? To me the -2oC sounds like a fairly conservative assumption already.

Line 448 ‘the time resolution of LPX’ This sounds indeed too coarse. Wouldn’t the current implementation cause a low bias, because it corrects LPX when it is supposed to be frozen, while it doesn’t do anything when it is supposed to have thawed? In the end, a sizeable portion of it could simply be explained by a limiting time resolution.   

Figure 3: It should be explained clearer what is shown here. The difference between LPX FT and LPX suggests LPX FT – LPX. However, then negative values would mean an improvement in e.g. RMSE. Given that the caption mentions the opposite it is probably the other way around. In case of the bias, it is even more confusing as a bias can be positive and negative. 

How is the bias defined? I would guess (LPX – Obs) – (LPX FT – Obs). However, in this case the Obs drop out of the equation. What remains is LPX – LPX FT, for which it is impossible to tell from the sign whether the F/T estimates have improved or not. As explained in the text LPX and LPX Ft refer to simulations of the LPX model. It calculates fluxes not mixing ratios, whereas comparisons that are shown here are in unit of ppb. This should be clarified.

Table 2, To be able to judge the significance of these differences the posterior flux uncertainty needs to be added to this table.  

Figure 5: Do LPX and LPX FT include all a priori emissions in CTE, or only the contribution of biospheric fluxes from LPX?

Figure 6: Unlike it is suggested in the caption, the color scale of figure b is about the same as figure a. I don’t see the point about figure b being an “order of magnitude larger than in the other figures”.    

Figures 7-9: In the lower panels of these figures, it is difficult to see the differences between the partly overlapping lines. When overlap occurs, it is not clear which colors overlap. A solution is needed to solve this (some colors in dashed lines?)

Figure 9: What happens in the winter of 2014 in CTE?

TECHNICAL CORRECTIONS

Line 20, ‘over A quarter’

Line 62, ‘Other’ i.o. ‘Another’;  ‘.. are ..’ i.o. ‘is’

Line 64, ‘Assimilate’ i.o. ‘Assimilates’

Line 80, ‘..the European …’

Line 83,’..has..’ i.o. ‘..is..’

Line 170: ‘Methane fluxes are … at their minimum …’

Line 174: ‘.. from now on …’

Line 176: ‘…called from now on …’

Line 180: ‘… when the fraction of frozen pixels was at least 90%’?

Line 184: ‘.. definitions which describe ..’

Line 268, ‘.. a couple of months ..‘

Line 276, ‘.. a couple of months ..‘

Line 299,’..most improved..’

Line 304, ‘sites provides’ note plural / singular

Line 309, ‘compared within the NHL sites’ What is meant here?

Line 316, ‘… the effect of SMOS F/T on the inversion…’

Line 317, ’observed’ i.o. ‘observation’

Author Response

We thank the reviewer for their nice overview and thorough comments. Please, find our answer in the attached file.

Reviewer 2 Report

Review of “Utilizing Earth Observations of Soil Freeze/Thaw data and Atmospheric Concentrations to Estimate Cold Season Methane Emissions in the Northern Hemisphere” written by Tenkanen et al

The manuscript investigates the impact of improved prior biospheric methane fluxes on the optimised methane fluxes obtained by an inverse modelling system, in particular for the northern high latitude. To obtain the improved biospheric methane emission inputs from LPX-Bern DYPTOP model estimates, remote sensing of freeze/thaw status from the SMOS satellite dataset is used to define more accurate freezing and thawing timing. Indeed, this idea works as the improvement in the prior flux can be seen by modelled CH4 mole fractions to some extent. Also, the manuscript is well organized and easy to follow. However, evidence from the inversion experiments to support the author’s argument is weak. Therefore, modifying and adding some essential factors is required to deliver better meaningful findings. Specific concerns can be found in the following comments.

  1. L201: is the average improvement statistically significant? Because the improvement is too small (just a few ppb) compared to the difference between observed and modelled prior mole fractions (tens of ppb). This question is also valid for the comparison between CTE-FT and CTE. Are those optimized fluxes, especially in the wintertime, different significantly?
  2. Throughout the manuscript, no uncertainty of optimized fluxes is presented (e.g. Table 2 and the main text). Does CTE estimate the uncertainty of the optimized fluxes?
  3. It looks that Figure 5 is not appropriate to represent what the authors are arguing for:
    • the legend is blocking the time series in some panels. Thus, the size and location of the legend should be modified.
    • the differences between LPX vs. LPX FT and CTE vs. CTE FT are not shown clearly. The authors need to come up with an idea to represent the differences more clearly. For example, it could be possible to replace this figure by a Taylor diagram and move the original figure into the supplementary.
    • for double-checking, are data shown daily average? The figure caption does not say it explicitly.
    • the rejection criteria are 30 ppb as explained in L130-131. But looking at figure 5, the difference between prior and posterior modelled CH4 mole fractions look far larger than 30 ppb. Were they all rejected in the inversion? Please clarify this.
  4. In addition, the prior modelled CH4 mole fractions at these six sites in Figure 5 looks too low compared to the actual observed CH4 mole fractions. Does this happen only at these sites or also happen at other sites used in the inversion? From the figure, the IC of CH4 looks too low. Maybe it explains the small improvement in prior fluxes measured in the observation space. How was the IC obtained? Also, if the IC is too low, is it possible to increase the level of IC to match with the observed values to simulate better results?
  5. L412-421: Is this interference the reason for no F/T region over central Asia and Mongolia as shown in Figure 1?
  6. Table 2: Missing uncertainty from the inversion experiment: how the different of optimizes fluxes from CTE vs. CTE-FT, are they statistically significant? Are the optimized fluxes from two experiments within the range of uncertainty or above/below the uncertainty, especially in the wintertime?

Minor concerns:

  1. Maybe the intermediate (2x3) zoom regions of the TM5 model exists around the 1x1 domain?
  2. L103: what does obvious erroneous soil F/T estimates mean by? Could you add a brief explanation about this?
  3. L135: Figure 3 is referred to earlier than Figures 1 and 2, which does not look organized.
  4. Table 1: The definition of abbreviation for institutes are missing in this table as well as Table S1. Adding the definition in the caption would be helpful.
  5. L180: period is missing at the end of the sentence.
  6. L204: it could be November 6 & April 30, for example, instead of 6.11. and 30.4. for readability.
  7. Figure 1: For panels a and b, the legend is not necessary to cross 0 days of the year. It could confuse as freezing happens before the end of a year and thawing happens after the start of a year.
  8. L219: “of” is duplicated at the end.
  9. L300-312: This paragraph highlights the importance of a specific site (Baranova, Russia). However, no data is referred to support this sentence. Probably some results are already shown in Figures 3 and 4. But it is very difficult to find where it is from the figures.
  10. L327: please change ppd to ppb.
  11. L348: is this outlier observation rejected or assimilated in the inversion?
  12. L380-383: Also, it could be partially due to the temporal optimization resolution of estimated fluxes by the inversion system. Isn’t it?
  13. L516-517: Please present the uncertainty range for the optimized fluxes.
  14. Figure S2: The figure is hard to understand due to the missing legend. What do boxes with dashed lines mean? it is unclear where R1, R2 and R3 are located and what colours mean, etc.
  15. Table S2: Probably values in the bracket to indicate the LPX-LPX FT are missing for second and third lines in the table (both top and bottom), compared with Table 2 in the main text.

Reviewer 3 Report

The manuscript presents interesting results aimed to improve an estimation of methane emission in the northern high latitudes. Also, the results can be useful for validation of land-surface models.

Comments:

Conclusions of the authors are based on in situ measurements of methane mole fraction. Were the results compared with satellite data of methane measurements, which have large coverage?

The section 2.4 “Definitions” is placed in the text after the sections where the definitions were already used. 

Page 8. Additional explanation is needed about detrended data. Why was linear regression used?

The paper can be accepted after corrections.

Reviewer 4 Report

See the attached comments.

Round 2

Reviewer 2 Report

I appreciate authors for their efforts to improve the quality of the manuscript. Authors addressed most comments raised by the reviewer thoroughly. I have only minor comments on the editorial issues. Please pay attention to the editorial issues in the manuscript.

L75: define “(CTE-CH4)” here not in L138

L205: “lags” maybe “weeks” is more accurate?

L206: period 2010-2018 here is confusing because author responded in the cover letter that CTE-CH4 model run starts in 2008.

Full names for NOAA/GML and MGO are missing in the manuscript.

Figures S2: What is the difference between panel a and b? The figure caption does not explain what they are separately.

Author Response

We thank again the reviewer for the comments. We have now modified the manuscript according to the comments:

L75: define “(CTE-CH4)” here not in L138

Defined now earlier as requested.

L205: “lags” maybe “weeks” is more accurate?

Yes, with lags we mean weeks. This is now changed.

L206: period 2010-2018 here is confusing because author responded in the cover letter that CTE-CH4 model run starts in 2008.

We modified the sentence in L206 to be: The simulations were conducted for the years 2010-2018 with 2 years (2008-2009) spin-up with LPX as a biospheric a priori.

Full names for NOAA/GML and MGO are missing in the manuscript.

They are now defined, NOAA/GML in Section 2.3 and MGO in Table 1.

Figures S2: What is the difference between panel a and b? The figure caption does not explain what they are separately.

Panel a is the optimisation regions for biospheric sources and panel b for anthropogenic. This is now defined in the figure's caption.

The manuscript will also be sended to a English editing service.